



# Turbulence characterization from a forward-looking nacelle lidar

Alfredo Peña[1], Jakob Mann[1], and Nikolay Dimitrov[1]

[1]DTU Wind Energy, Technical University of Denmark, Roskilde, Denmark

*Correspondence to:* Alfredo Peña (aldi@dtu.dk)

**Abstract.** We present two methods to characterize turbulence in the turbine inflow using radial velocity measurements from nacelle-mounted lidars. The first uses a model of the three-dimensional spectral velocity tensor combined with a model of the spatial radial velocity averaging of the lidars and the second uses the ensemble-averaged Doppler radial velocity spectrum. With the former, filtered turbulence estimates can be predicted, whereas the latter model-free method allows us to estimate unfiltered turbulence measures. Two types of forward-looking nacelle lidars are investigated: a pulsed system that uses a 5-beam configuration and a continuous-wave system that scans conically. For both types of lidars, we show how the radial velocity spectra of the lidar beams are influenced by turbulence characteristics and how to extract the velocity-tensor parameters that are useful to predict the loads on a turbine. We also show how the velocity-component variances and co-variances can be estimated from the radial-velocity unfiltered variances of the lidar beams. We demonstrate the methods using measurements from an experiment conducted at the Nørrekær Enge wind farm in northern Denmark, where both types of lidars were installed on the nacelle of a wind turbine. Comparison of the lidar-based along-wind unfiltered variances with those from a cup anemometer installed on a meteorological mast close to the turbine shows a bias of just 2%. The ratios of the unfiltered and filtered radial velocity variances of the lidar beams to the cup-anemometer variances are well predicted by the spectral model. However, other lidar-derived estimates of velocity-component variances and co-variances do not agree with those from a sonic anemometer on the mast, which we mostly attribute to the small cone angle of the lidar. The velocity-tensor parameters derived from sonic-anemometer velocity spectra and those derived from lidar radial velocity spectra agree well under both near-neutral atmospheric stability and high wind-speed conditions with differences increasing with decreasing wind-speed and increasing stability. We also partly attribute these differences to the lidar beam configuration.

## 1 Introduction

Recently, lidars have been mounted on the nacelle of wind turbines to investigate wake characteristics (Bingöl et al., 2010; Machefaux et al., 2016; Trujillo et al., 2016) and are nowadays extensively used in a forward-looking (FL) mode to scan the turbine inflow for many purposes. One of such is power-performance measurements; FL nacelle lidars decrease the statistical uncertainty of the measured power curve when compared to that based on mast measurements (Wagner et al., 2014). The statistical uncertainty associated with load validation can potentially also be reduced (Dimitrov and Natarajan, 2016). Another important use of FL nacelle lidars is turbine control; they have the potential to reduce loads and increase energy capture (Mikkelsen et al., 2013; Schlipf et al., 2015). Irrespectively of the application, FL nacelle lidars are primarily aimed to



characterize the inflow in front of the turbine. Inflow characterization has been performed using lidars of different types and configurations for several years (Hardesty et al., 1981; Peña et al., 2010; Aitken et al., 2012). However, FL nacelle lidars have the advantage of measuring the inflow in front of the turbines more 'effectively' than other types of lidars because they scan over the area in front of the turbine and yaw with it. Therefore, they can potentially be used for measuring the yaw misalign-
ment of wind turbines (Fleming et al., 2014). If they become widely applied in the wind-energy industry, they could be used to characterize wind resources in regions where measurements from meteorological towers are scarce or non-existent.

Similar to ground-based lidars, there are two main types of FL nacelle lidars, pulsed and continous-wave (CW), which mainly differ, for the purpose of turbulence estimation, on the measurement probe volume and the scanning strategies (specific details are given later). As with any other Doppler lidar, they only measure the radial velocity along the laser beam or line-of-
sight velocity. As their measurement probe volumes are generally larger than those of cup and sonic anemometers, they might not be able to measure small turbulent eddies, which leads to 'filtered' turbulence estimates but as they scan the atmosphere with laser beams in different directions, there might be contributions (contamination) from different velocity components that can lead, for some scanning configurations and under certain turbulence conditions, to turbulence estimates that might be even higher than those from cup or sonic anemometers. A detailed analysis on how lidar-based turbulence estimates can be assessed,
filtered, and contaminated is presented in Sathe and Mann (2013).

Here we use time series of radial velocity measurements from different beams emitted by a FL nacelle lidar to estimate the turbulence parameters of the three-dimensional spectral velocity tensor model by Mann (1994) (hereafter the Mann model). This model is chosen because it fits well the atmospheric-turbulence velocity spectra for different surface, wind, and atmospheric-stability conditions within the first ≈100 m from the ground (Peña et al., 2010; Chougule et al., 2015) and
is widely used to perform aerolastic simulations of wind turbines. The ultimate objective of this study is to find out if nacelle lidars can be used independently (i.e. without the need of extra measurements, e.g. from instruments on meteorological masts) to extract turbulence information from the inflow. Nacelle lidars can potentially infer the inflow characteristics that actually impact the turbines better than traditional nacelle or mast anemometry because they can scan over an air volume, which is more representative of the flow entering the rotor plane. We also use, when possible, information of the Doppler radial velocity
spectrum to estimate the 'unfiltered' lidar beam variances and, from those, we estimate the velocity-component variances and co-variances (Mann et al., 2010).

This paper is organized as follows. In Sect. 2, we introduce shortly the characteristics of the wind field, how this is represented by the Mann model, and how to extract the turbulence characteristics from velocity spectra. Section 3 shows the two types of nacelle lidars investigated here, Sect. 3.1 illustrates how to derive the radial velocity spectra from the different lidar
configurations, how these spectra are influenced by both the lidar configuration and the turbulence characteristics of the Mann model, and, Sect. 3.2 how to extract turbulence information from the lidars' radial velocity spectra. Section 4 introduces the Nørrekær Enge wind farm and the measurements of the experimental campaign, Sect. 5 provides the details on the way the measurements are analyzed, and Sect. 6 shows the comparison of turbulence characteristics extracted from nacelle-lidar measurements and those from sonic- and cup-anemometer measurements. Finally, we provide some discussion and conclusions in
the last two sections.





## 2 General background

The wind field is described by a vector field $\boldsymbol{u}(\boldsymbol{x})$, where the time argument is eliminated because Taylor's frozen turbulence hypothesis is assumed (Mizuno and Panofsky, 1975) and $\boldsymbol{x}$ is the position vector in space, $\boldsymbol{x} = (x, y, z)$. The mean value of the homogeneous velocity field is $\langle \boldsymbol{u}(\boldsymbol{x}) \rangle = (U, 0, 0)$, so the coordinate $x$ is in the mean wind direction. In other words, the wind field as a function of space and time can be written as

$$\boldsymbol{u}(\boldsymbol{x}, t) = \boldsymbol{u}(x - Ut, y, z). \tag{1}$$

The wind field can also be written as a Fourier integral,

$$\boldsymbol{u}(\boldsymbol{x}) = \int \boldsymbol{u}(\boldsymbol{k}) \mathrm{e}^{\mathrm{i}\boldsymbol{k}\cdot\boldsymbol{x}} d\boldsymbol{k} \Leftrightarrow \boldsymbol{u}(\boldsymbol{k}) = \frac{1}{(2\pi)^3} \int \boldsymbol{u}(\boldsymbol{x}) \mathrm{e}^{-\mathrm{i}\boldsymbol{k}\cdot\boldsymbol{x}} d\boldsymbol{x}, \tag{2}$$

where $\boldsymbol{k}$ is the wave vector and we are a bit slack and do not use the more rigorous Fourier-Stieltjes notation (Batchelor, 1953; Mann, 1994). The ensemble average of the absolute squared Fourier coefficients is the spectral tensor:

$$\langle u_i^*(\boldsymbol{k}) u_j(\boldsymbol{k'}) \rangle = \Phi_{ij}(\boldsymbol{k}) \delta(\boldsymbol{k} - \boldsymbol{k'}) \tag{3}$$

and it is, according to the Wiener-Khinchin theorem, related to the covariance tensor $R_{ij}(\boldsymbol{r}) \equiv \langle u_i(\boldsymbol{x}) u_j(\boldsymbol{x} + \boldsymbol{r}) \rangle$, where $\boldsymbol{r}$ is the separation vector, by

$$\Phi_{ij}(\boldsymbol{k}) = \frac{1}{(2\pi)^3} \int R_{ij}(\boldsymbol{x}) \mathrm{e}^{-\mathrm{i}\boldsymbol{k}\cdot\boldsymbol{x}} d\boldsymbol{x}. \tag{4}$$

Since $\boldsymbol{u}^*(\boldsymbol{k}) = \boldsymbol{u}(-\boldsymbol{k})$, Eqn. (3) can also be written as

$$\langle u_i(\boldsymbol{k}) u_j(\boldsymbol{k'}) \rangle = \Phi_{ij}(\boldsymbol{k}) \delta(\boldsymbol{k} + \boldsymbol{k'}). \tag{5}$$

### 2.1 Turbulence characterisation

The spectral velocity tensor $\Phi_{ij}$ is assumed to be described by the Mann model, which, besides $\boldsymbol{k}$, only contains three parameters: $\alpha\varepsilon^{2/3}$, $L$, and $\Gamma$, where $\alpha$ is the spectral Kolmogorov constant, $\varepsilon$ the specific rate of destruction of turbulent kinetic energy, $L$ a length scale related to the size of the turbulent eddies, and $\Gamma$ a parameter describing the anisotropy of the turbulence.

From the spectral tensor, the one-point spectra are calculated by

$$F_{ij}(k_1) = \iint \Phi_{ij}(\boldsymbol{k}) dk_2 dk_3 \tag{6}$$

and, typically, the three auto-spectra of the $u$-, $v$-, and $w$-components of the wind velocity, $F_{11}$, $F_{22}$, and $F_{33}$, respectively, together with the one-point cross-spectrum $F_{13}$ are fitted simultaneously to measured or theoretical spectra in order to obtain the Mann-model parameters (hereafter referred to as Mann parameters). This procedure is described in Mann (1994). In order to facilitate the fitting, a two-parameter look-up table (LUT) with values of $F_{ij}(k_1) = F_{ij}(k_1; \alpha\varepsilon^{2/3} = 1, L = 1, \Gamma)$ is precomputed. The mathematical identity

$$F_{ij}(k_1; \alpha\varepsilon^{2/3}, L, \Gamma) = L^{5/3} \alpha\varepsilon^{2/3} F_{ij}(k_1 L; 1, 1, \Gamma) \tag{7}$$

is used to calculate the spectra for arbitrary values of $k_1$, $\alpha\varepsilon^{2/3}$, $L$, and $\Gamma$.



## 3 Nacelle lidars

Two types of FL nacelle lidars are investigated: a CW and a pulsed lidar. The lidars are assumed to be mounted close to the center of the rotor with $N$ beams pointing in different directions (see Fig. 1). For the CW lidar studied here, the beams point on a cone with the symmetry axis pointing upstream (Fig. 1-left shows a configuration with $N = 13$, of which 12 beams draw

a conical surface and one is perpendicular to the rotor plane). The half opening angle of the cone is $\varphi$. The beams of the CW lidar are focused at some distance $d_f$. The lidar is not focused at a point but rather in a pencil thin, many meters long volume (Rodrigo and Pedersen, 2012; Angelou et al., 2012). In case other measurement planes are required, refocusing of the laser beam is necessary. For the pulsed lidar studied here ($N = 5$), the beam directions also form a cone where four positions are within the conical surface and one is perpendicular to the rotor plane (Fig. 1-right).

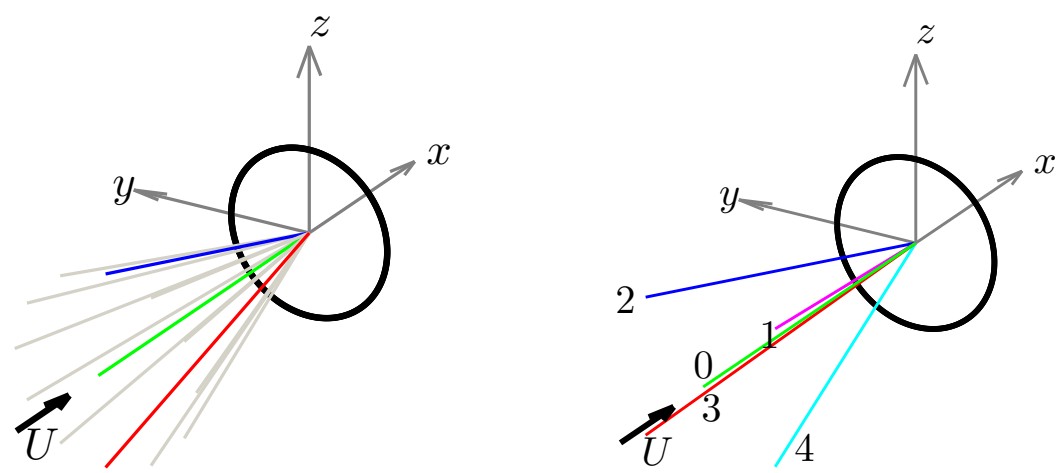

**Figure 1.** Geometry of the rotor and nacelle lidars. The $x$-axis is in the mean wind direction. The lidar beams point upwind in the directions determined by the unit vectors $\boldsymbol{n}_i$. For the CW lidar (left frame) we include a beam perpendicular to the rotor for comparison only. For the pulsed lidar (right frame) we show a 5-beam configuration where beam 0 is perpendicular to the rotor

The $i$th lidar beam points to the direction defined by the unit vector $\boldsymbol{n}_i$ ($i = 1, ..., N$). The first coordinate $n_{i1}$ of $\boldsymbol{n}_i$ will always be negative because the beams are pointing upwind of the rotor. The unit vector can be expressed as

$$\boldsymbol{n} = (-\cos\varphi, \sin\varphi\cos\theta, \sin\varphi\sin\theta), \tag{8}$$

where $\theta$ is the angle between the $y$-axis and $\boldsymbol{n}$ projected onto the $y$-$z$-plane. The unit vector of the beam perpendicular to the rotor is $\boldsymbol{n} = (-1, 0, 0)$. If we assume that the lidars measure at a point, instead of over a probe volume, and that the $u$-, $v$-,

and $w$-components do not change over the scanned area, the radial velocity of the lidar beams over the scanned circle can be estimated as

$$v_r(\theta) = -u\cos\varphi + v\sin\varphi\cos\theta + w\sin\varphi\sin\theta. \tag{9}$$



### 3.1 Lidar radial velocity spectra and beam variances

Considering the lidar probe volume and the scanning configuration, the radial velocity measured by the $i$th lidar beam may be approximated by

$$v_{r_i}(\boldsymbol{x}) = \int_{-\infty}^{\infty} \boldsymbol{n}_i \cdot \boldsymbol{u}(s\boldsymbol{n}_i + \boldsymbol{x})\phi(s - d_f)ds, \tag{10}$$

where the argument $\boldsymbol{x}$ is the position of the lidar with a first component decreasing with time as $-Ut$ using Taylor's hypothesis and $\phi$ is the lidar's weighting function that considers the probe volume. For a CW lidar, it is typically approximated by

$$\phi(s) = \frac{1}{\pi} \frac{z_R}{z_R^2 + s^2}, \tag{11}$$

where $z_R$ is the Rayleigh length (Sonnenschein and Horrigan, 1971) that can be estimated as

$$z_R = \frac{\lambda d_f^2}{\pi r_b^2}, \tag{12}$$

where $\lambda$ is the laser wavelength and $r_b$ the beam radius at the output lens.

For the pulsed lidar, the weighting function is approximated by

$$\phi(s) = \frac{z_R - |s|}{z_R^2}, \tag{13}$$

for $|s| < z_R$ and $\phi(s) = 0$ elsewhere. $z_R$ in Eqn. (13) is not the Rayleigh length as in Eqn. (11) but half the length of a rectangular pulse (Mann et al., 2009). Despite this discrepancy, we use the same symbol because $z_R$ is the parameter that

characterizes both weighting functions.

For lidar measurements, two complications arise when deriving velocity spectra. Firstly, we do not obtain measurements of the velocity components individually, as with a sonic anemometer, but certain fixed combinations of components. Secondly, the measurements are not obtained in one point, but rather a weighted average as expressed in Eqn. (10). The strategy is to calculate theoretical spectra that include both the effect of pointing the lidar in the direction $\boldsymbol{n}_i$ and of averaging. Then, the

measured spectra are fitted to the theoretical to get the turbulence parameters. One can expect this procedure to be unsuccessful for $z_R > L$, i.e. if the lidar is averaging out most eddies.

The first step is to derive an expression for the spectra measured by a single beam of the lidar. Following the steps given in Mann et al. (2009) and Sjöholm et al. (2009), the result is

$$F_{v_r}(k_1) = n_i n_j \iint \left|\hat{\phi}(\boldsymbol{k} \cdot \boldsymbol{n})\right|^2 \Phi_{ij}(\boldsymbol{k})dk_2 dk_3, \tag{14}$$

where $\hat{\phi}$ is the Fourier transform of the weighting function $\phi$. For the CW lidar this is an exponential function,

$$\hat{\phi}(k_1) = \exp(-|k|z_R), \tag{15}$$

and

$$\hat{\phi}(k_1) = \text{sinc}^2(kz_R/2), \tag{16}$$



for the pulsed lidar. Notice that $F_{v_r}$ is not a function of $d_f/L$ because the turbulence is assumed homogeneous.

Examples of radial velocity spectra of the CW and pulsed lidars calculated from Eqn. (14) with a half opening angle of $\varphi = 15°$ compared with the 'ideal' sonic $u$-spectrum are shown in Figs. 2 and 3, respectively. It is seen that the beam pointing most upward generally has the most variance. This is expected because of the negative correlation between the horizontal and vertical velocity components. The difference between the downward and upward pointing beam spectra is smaller than the differences between ordinary velocity-component spectra and deteriorates with increasing $z_R/L$. We can also see that for the pulsed lidar the radial spectra of the top beams (1 and 2) are above the sonic spectrum for $z_R/L = 0.25$, which is due to contributions from different components of the spectral tensor. Similar mechanisms can result in a middle beam radial velocity spectrum above the the top beam ones, particularly for $z_R/L \geq 1$.

Figure 4-left shows the behavior of the ratio of the lidar beam radial velocity variance, $\sigma_{v_r}^2$, to the variance of the $u$-component, $\sigma_u^2$, for a number of $z_R/L$ values and for both types of lidars based on the Mann model with $\Gamma = 3$. As expected from the results in Figs. 2 and 3, the ratio increases with decreasing $z_R/L$ and for the 0/middle beam of the pulsed/CW lidars, $\sigma_{v_r}^2/\sigma_u^2 = 1$ at $z_R/L = 0$ as no averaging due to probe volume occurs. Further, both lidars' top beams variances can be higher than $\sigma_u^2$ for $z_R/L \approx 0$. Another way to study the contributions of the different velocity components to $\sigma_{v_r}^2$ is shown in Fig. 4-right. There we illustrate the ratio of the variance of the other two components, $\sigma_v^2$ and $\sigma_w^2$, as well as $\sigma_u^2$ to $\sigma_{v_r}^2$ as function of the beam azimuthal position for $z_R/L = 0$. With $\varphi = 15°$ and such turbulence characteristics, we can only measure a portion of $\sigma_v^2$ and $\sigma_w^2$, and $\sigma_{v_r}^2 \approx \sigma_u^2$ at $\theta \approx 11°/169°$ (also if a middle beam is used no matter the turbulence characteristics). For the same turbulence characteristics as those used in Fig. 4, if we use a lidar with $\varphi = 60°$, $\sigma_{v_r}^2 < \sigma_u^2$ for all azimuthal positions, whereas $\sigma_{v_r}^2 \approx \sigma_v^2$ at $\theta \approx 200°/340°$ and $\sigma_{v_r}^2 \approx \sigma_w^2$ at $\theta \approx 237°/303°$ (not shown). It is also observed that for the same $z_R/L$ value, the averaging by the CW lidar has a stronger effect on the variance than the pulsed lidar.

### 3.1.1 Unfiltered lidar radial velocity variance

The unfiltered variance of the lidar beams, $\sigma_{v_{r,\text{unf}}}^2$, can be estimated by using the information of the instantaneous Doppler radial velocity spectrum. Following the steps in Mann et al. (2010) or Branlard et al. (2013), the ensemble-average Doppler spectrum of the radial velocity $\langle S(v_r) \rangle$ can be assumed to be equal to the probability density function of $v_r$, i.e. $\langle S(v_r) \rangle = p(v_r)$. This is because the average of $v_r(s)$ does not change highly with $s$, as FL nacelle lidars use a small cone angle and so the velocity gradient along the probe volume is negligible. Therefore, $\sigma_{v_{r,\text{unf}}}^2$ can be estimated as the second central moment of $p(v_r)$.

Once $\sigma_{v_{r,\text{unf}}}^2$ is computed and assuming homogeneous turbulence, the scanning pattern can be used to extract the velocity-component variances by taking the variance of $v_r$ in Eqn. (9),

$$\sigma_{v_{r,\text{unf}}}^2(\theta) = \sigma_u^2 \cos^2\varphi + \sigma_v^2 \sin^2\varphi \cos^2\theta + \sigma_w^2 \sin^2\varphi \sin^2\theta - 2\overline{u'w'}\cos\varphi \sin\varphi \sin\theta, \qquad (17)$$

where $\overline{u'w'}$ is the $uw$-covariance, the primes denote fluctuations, the overbar a time average, and we ignore the terms where $\overline{u'v'}$ and $\overline{v'w'}$ appear because these two are usually small. In case of misalignment of the lidar beams with respect to the wind, either because of misalignment of the turbine with the wind (yaw misalignment), lidar misalignment with the turbine, or both, it is not difficult to derive an expression for $\sigma_{v_{r,\text{unf}}}^2$ that accounts for the misalignment angle $\beta$.



**Figure 2.** Sonic and CW lidar velocity spectra from Eqns. (6) and (14) corresponding to the beams shown in Fig. 1-left. Values of $z_R/L$ are indicated, $\varphi = 15°$, $\Gamma = 3$, and $\alpha\varepsilon^{2/3} = 0.1$ m$^{4/3}$ s$^{-2}$







**Figure 3.** Sonic and pulsed lidar velocity spectra from Eqns. (6) and (14) corresponding to the beams shown in Fig. 1-right. Values of $z_R/L$ are indicated, $\varphi = 15°$, $\Gamma = 3$, and $\alpha\varepsilon^{2/3} = 0.1$ m$^{4/3}$ s$^{-2}$





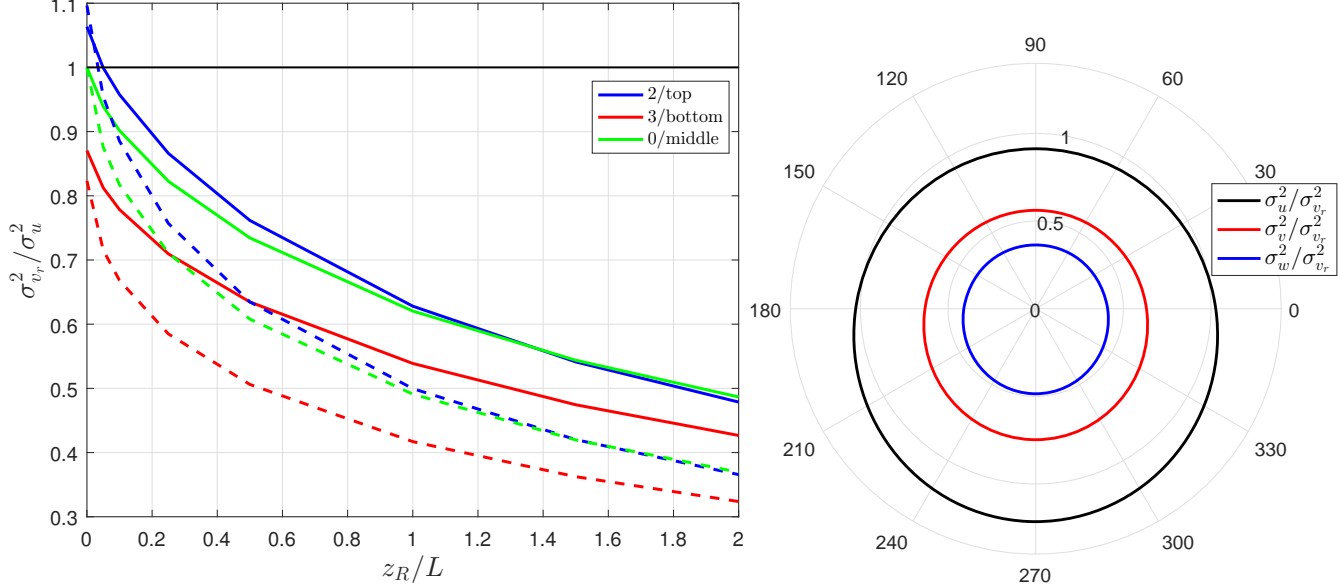

**Figure 4.** (Left frame) variance of the lidar beams' radial velocity (divided by the $u$-component variance) of the 2/top, 3/bottom and 0/middle beams of the pulsed/CW (solid/dashed lines) lidars as a function of $z_R/L$. (Right frame) variance of each of the velocity components (divided by the lidar beam's radial velocity) as function of azimuthal position. Turbulence characteristics are computed for $\varphi = 15°$ and $\Gamma = 3$

### 3.2 Turbulence characterisation from nacelle-lidar measurements

The computational burden of creating a lidar-based LUT using Eqn. (14) is larger than in the standard case, i.e. using Eqn. (7), because the lidar radial velocity spectrum is not only a function of the two parameters $k_1 L$ and $\Gamma$, but also $z_R/L$, $\varphi$, and $\theta$. Further, lidar beam misalignment can be an issue. Therefore, we need to add an extra dimension to the LUT because such
5  misalignment has a large effect on the lidar radial velocity spectrum.

Figure 5 illustrates the effect of misalignment ($\beta = -2°$) on the pulsed lidar radial velocity spectra for a set of Mann parameters. The effect of the relatively small misalignment is noticeable; the spectrum of the beams that become more parallel to the wind is clearly above that of those becoming less parallel at the same height. For this particular pulsed lidar configuration, misalignment can result in a similar spectrum for the beams 0 (middle) and 1/2 (depending on the sign of the misalignment).

10  ## 4 Site and measurements

### 4.1 Site

The Nørrekær Enge wind farm is located in the Himmerland region in northern Jutland, Denmark, ≈300–400 m south-east of the waters of Limfjorden (see Fig. 6). It comprises 13 Siemens 2.3 MW-93 wind turbines with hub height of 81.8 m and a





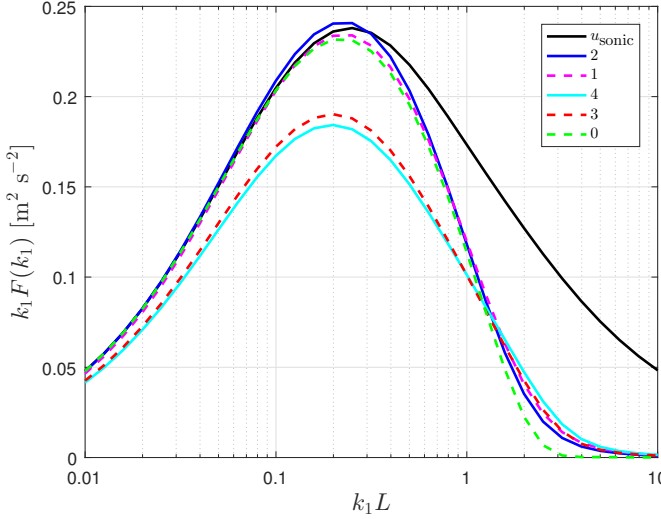

**Figure 5.** Effect of lidar beam misalignment (with respect to the wind) on the radial velocity spectra of a pulsed lidar for $\varphi = 15°$, $\Gamma = 3$, $\alpha\varepsilon^{2/3} = 0.1$ m$^{4/3}$ s$^{-2}$, $z_R/L = 0.5$, and $\beta = -2°$

rotor diameter $D$ of 92.6 m. They are aligned on a row at a direction $73.9°$ with the north. The distance between the turbines is 487 m ($5.2D$). A meteorological mast was located $101.2°$ at a distance of 232 m ($2.5D$) from turbine number 4 (from left to right on the row). The wind farm is located over flat terrain and the surface is characterized by a mix between croplands and grasslands, and the fjord to the north. At $\approx2$ km south-west of turbine 4, the terrain is not longer flat.

5 ## 4.2 Measurements

The measurements here analyzed correspond to the period October 27, 2015 to January 7, 2016. There are three types of measurements: supervisory control and data acquisition (SCADA) on turbine 4, FL nacelle-lidar measurements from systems mounted on the nacelle of turbine 4, and meteorological mast observations. Both lidars were pre-tilted down $\approx0.30°$ so that their axes pointed at hub height at a position $2.5D$ from the turbine for maximum power-performance operating conditions,

10 based on aerolastic simulations of the tower bending (Andrea Vignaroli, personal communication).

### 4.2.1 Turbine measurements

For this analysis we use the following SCADA 10-min means of turbine 4: yaw, power, and turbine and grid status. The yaw and power signals provide measurements of the position of the turbine and the converted power, and the grid and turbine status signals show whether the turbine was grid-connected (yes/no) and available (yes/no).



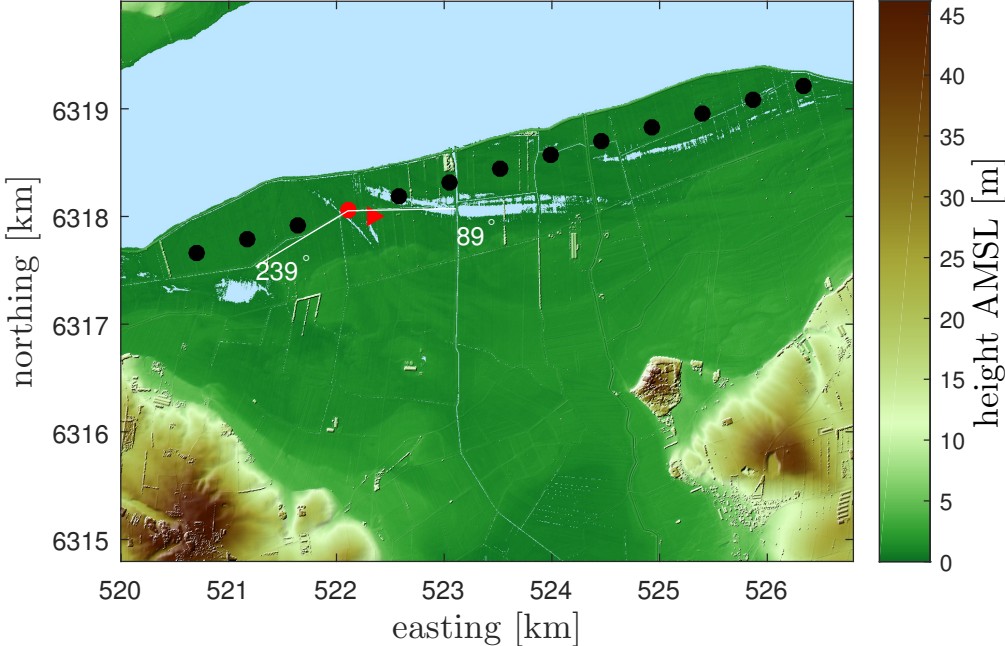

**Figure 6.** The Nørrekær Enge wind farm in northern Denmark on a digital surface elevation model (UTM32 WGS84). The wind turbines are shown in circles, that with the nacelle lidars in red and the mast in a triangle. The sector used for the analysis is also indicated. The waters of Limfjorden are shown in light blue

### 4.2.2 Pulsed lidar

A 5-beam Avent pulsed lidar (hereafter known as Avent) was mounted on the nacelle of turbine 4. 10 different ranges were measured simultaneously per beam position (49, 72, 95, 109, 121, 142, 165, 188, 235, and 281 m). The beam configuration is exactly as that in Fig. 1-right frame (and we will use the same beam numbering) with $\varphi = 15.08°$ and $z_R = 24.75$ m (Borraccino et al., 2015). The lidar accumulated radial velocity spectra per beam position during 1 s before it moved to the next beam position; thus, radial velocity time series can be analyzed at 0.2 Hz. Each radial velocity estimate from the average Doppler spectrum was performed by the instrument using a maximum-likelihood-estimator algorithm (Peña et al., 2015).

### 4.2.3 Continuous-wave lidar

A ZephIR dual-mode CW lidar (hereafter known as ZephIR) was also mounted on the nacelle of turbine 4. Five different ranges were considered (10, 30, 95, 120, and 235 m); for each range ≈50 azimuthal positions on the circle formed with a cone with $\varphi = 15.05°$ were measured during 1 s; the system averaged Doppler radial velocity spectra within azimuthal ranges of ≈7.38° to get an estimate of the radial velocity per azimuth by computing the centroid of the average Doppler spectrum (Borraccino et al., 2015). The system also kept a record of each average Doppler radial velocity spectrum, which is used here to estimate the





unfiltered variance. The lidar characteristics $\lambda = 1.56 \times 10^{-6}$ m and $r_b = 28$ mm (Michael Harris, personal communication) can be used to estimate $z_R$ with Eqn. (12). Each range was sampled three times before focusing to the next one; thus, radial velocities for the same range and azimuthal position can be found every $\approx$18 s.

### 4.2.4 Mast measurements

We use measurements from cup anemometers (P2546A) at 80, 78, and 57 m height mounted on 3-m long booms 250° from the north, from a sonic anemometer (CSAT3) at 76 m on a 2-m boom 190° from the north, and a wind vane (Vector W200P) at 78 m on a 3-m boom 70° from the north, all mounted on the meteorological mast. The mast is an equilateral triangular lattice structure with a width of 0.4 m at 80 m.

## 5 Data analysis

### 5.1 Data selection and filtering

We analyze the time series of all data and their statistics in 10-min periods. The total number of 10-min periods available for analysis is 9586; during the period of the campaign, five days were used to measure with the ZephIR at one range (235 m), thus we need to subtract 781 from the 10367 potential 10-min periods. The next steps are followed in the analysis:

1. We use the 10-min vane measurements to concentrate the analysis on a wake-free sector covering the mast location
(88.85°–238.85°) that takes into account the obliquity of the wind farm row and a 15° wake expansion (see Fig. 6). 5825 10-min are available for analysis where both lidars are also working (based on a 10-min status signal of both lidars) and turbine 4 is grid-connected and available.

2. The availability of the Avent data is highest at the range 121 m because this range is the closest to the focusing distance. Therefore, we focus all our lidar-data analysis at this range, although the mast is at 232 m from turbine 4. Further, when
a carrier-to-noise (CNR) filter is applied to the 5-s time series, the two lowest beams (3 and 4) return less data than the others due to i.a. obstruction from the blades (the availability of beam 3 is lower than that of beam 4). 3236 10-min periods are available for analysis after filtering the 5-s Avent data so that for each 10-min period there are a minimum of 110 samples for beams 0, 1, 2, and 4 with CNR $> -22$ dB.

3. We then extract all radial velocities for all azimuthal positions of the ZephIR for the range 120 m when no rain was de-
tected by the instrument. The azimuthal position of the $\approx$50 points over the scanned circle changes after each revolution. 2590 10-min are available for analysis in which there are a minimum of 4500 radial velocities samples per 10-min period at the 120-m range.

4. Finally, we extract the 1-Hz data of the sonic anemometer and cup anemometer at 80 m, in which there are a minimum of 600 samples per 10-min period. The final dataset thus contains 2273 10-min samples of concurrent turbine-lidars-mast
data.



Further, each 10-min time series has been post-processed. For the Avent data, we linearly detrend each radial velocity time series for each beam before applying a despiking filter where values above and below 3 standard deviations from the mean are filtered out. The missing values are then filled in using linear interpolation. Figure 7-top left shows an example of a 10-min time series of the Avent beams' radial velocity. The solid lines show the final interpolated time series and the markers the original radial velocities before post-processing.

For the ZephIR data, we construct time series of radial velocities at azimuthal positions similar to those of the Avent. Since the azimuthal positions of the ZephIR change from revolution to revolution, we extract radial velocities within azimuthal position bins of 7.2° on a fixed frame of reference. Three of such bins, 43, 6, and 31, are 'aligned' with the Avent beams 1, 2, and 4, respectively. The time series per bin is then threshold-filtered with a minimum radial velocity of 2 m s$^{-1}$, and detrended and despiked as with the Avent data. Figure 7-top right shows the time series per bin; we include 4 more bins (0, 12, 18, and 37) than those aligned with the Avent beams and their positions can be inferred by color-coding using Fig. 7-bottom left, which shows the radial velocities in a polar plot. In Fig. 7-top right, the effect of threshold-filtering and despiking is noticeable (the filtered time series are shown in solid lines and the original in markers) and in Fig. 7-bottom left all the radial velocities estimated from the Doppler spectrum within the 10-min period at the 120-m range by the ZephIR are shown. The plot clearly shows the effect of the blades on the measurements (the figure-of-eight close to zero radial velocity). In this latter plot, we also include the radial velocities of the three Avent beams that are aligned with the ZephIR bin positions. At these three positions, both lidars show good agreement; a comparison of all 10-min mean radial velocities estimated by the Avent and ZephIR for one of these 'aligned' positions, beam 2 and bin 6, respectively, is shown in Fig. 7-bottom right. Figure 7-top right also shows that it is possible to get more than one radial velocity value within the same azimuthal bin (sometimes up to three values). Finally, the ZephIR's time series are 'completed' using linear interpolation.

For each 10-min period, the 1-Hz sonic and cup anemometer data are detrended and despiked as with the lidar data, and mean and turbulence statistics are computed. The sonic-anemometer wind-speed components are rotated so that $u$ is aligned with the mean wind. We estimate the friction velocity, from the sonic wind-speed and temperature fluctuations, as

$$u_* = \left( \overline{u'w'}^2 + \overline{v'w'}^2 \right)^{1/4}, \tag{18}$$

and the Obukhov length estimated as,

$$L_O = -\frac{u_*^3}{\kappa(g/\overline{T})\overline{w'\Theta'_v}}, \tag{19}$$

where $\kappa$ is the von Kármán constant ($\approx 0.4$), $g$ the gravitational acceleration, $T$ a reference temperature, and $\Theta_v$ the virtual potential temperature. Spectra of all lidar radial velocities, sonic-anemometer wind-speed components and cup-anemometer horizontal wind velocity are computed for each 10-min period. All 10-min turbulence statistics and spectra from the sonic anemometer are also computed on a 5-s and a 18-s basis.





**Figure 7.** An example of a 10-min time series of the radial velocity of different beams for the Avent (top-left frame) and the ZephIR (top-right frame) lidar. The radial velocities of the two lidars at all azimuthal positions are illustrated in the bottom-left panel (see text for details). In the bottom-right panel, we show a comparison of the 2273 10-min mean radial velocities of the Avent (beam 2) and ZephIR (bin 6) with the results of a linear regression through the origin and coefficient of determination $R^2$





## 5.2 Sonic-anemometer measurements

When compared to the measurements from the 80-m cup anemometer, the sonic-anemometer mean horizontal wind speeds are 2.6% lower (see Fig. 8-left panel). This bias is higher than 0.6%, which is the estimation that results from assuming the logarithmic wind profile

$$U = \frac{u_*}{\kappa} \ln\left(\frac{z}{z_0}\right), \tag{20}$$

where $z_0$ is the roughness length, $\approx 0.012$ m, which is a typical value of these surface conditions (Peña et al., 2016). When looking at variances, the bias is 12% (Fig. 8-right panel), both if we use the $u$-component or the combined $u$- and $v$-components for the estimation of the sonic-anemometer variance. The latter means that for this site and at this height, the $v$-variance has a low contribution to the horizontal velocity variance (which is what a cup anemometer does theoretically measure) and so we could assume the cup-anemometer variance to give a good estimate of the $u$-variance. On the other hand, the bias between both instruments' variances cannot be explained simply; a 4% bias is expected assuming the 2% bias of the mean wind speed.

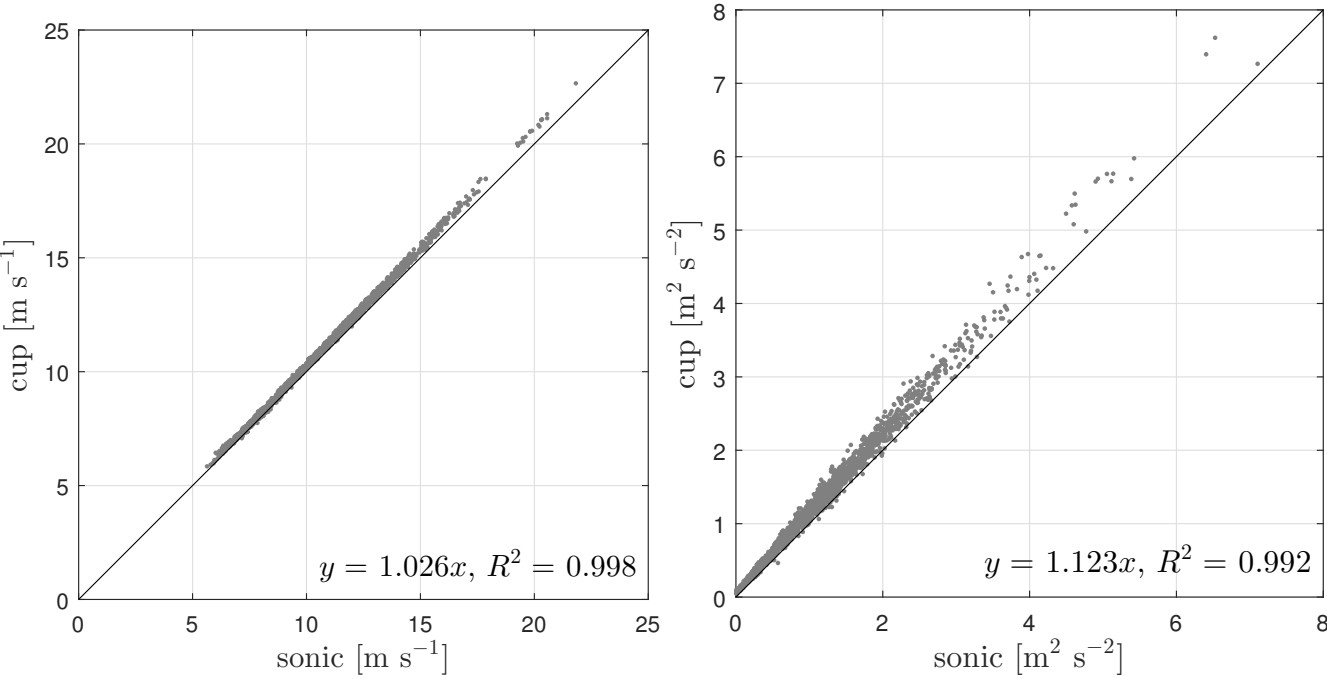

**Figure 8.** Comparison of sonic and 80-m cup anemometer statistics: mean wind speed (left frame) and horizontal wind variance (right frame). Each 10-min sample is shown in grey markers, a 1:1 line is shown for guidance in black, and the results of a linear regression through the origin and $R^2$ are given

The behavior of the sonic-derived velocity spectra does not correspond well with the notion of turbulence local isotropy within the inertial subrange, where we expect the same spectral density for the $v$- and $w$-components and the $u$-component





being 25% lower (Wyngaard, 2010). Figure 9 shows that within the inertial subrange the ensemble-average sonic $u$-spectrum (of all 10-min observed spectra) is indeed $\approx 25\%$ lower than the $v$-spectrum but so is the $w$-spectrum. Possible explanations of this are path-averaging errors and transducer shadowing mainly attenuating the $w$-spectrum measured by the CSAT3 (Horst and Oncley, 2006). Figure 9 also illustrates the fit to the three auto-spectra and cross-spectrum using the Mann model (see

Sect. 2.1), which shows the expected behavior within the inertial subrange. The fit is performed on the ensemble-average spectra that have been logarithmically-averaged on the basis of the wavenumber (we will use such logarithmically-averaged spectra when fitting Mann parameters). These 'average' Mann parameters ($\Gamma = 3.00$, $\alpha\varepsilon^{2/3} = 0.14$ m$^{4/3}$ s$^{-2}$, and $L = 35.38$ m) are similar to those observed at a site with similar surface and turbulence characteristics (Peña et al., 2010), but it should be noticed that these are the average of spectra for a number of atmospheric and turbulence conditions and that the Mann-model

fitting procedure is normally performed over specific wind-speed, turbulence, or atmospheric-stability ranges.

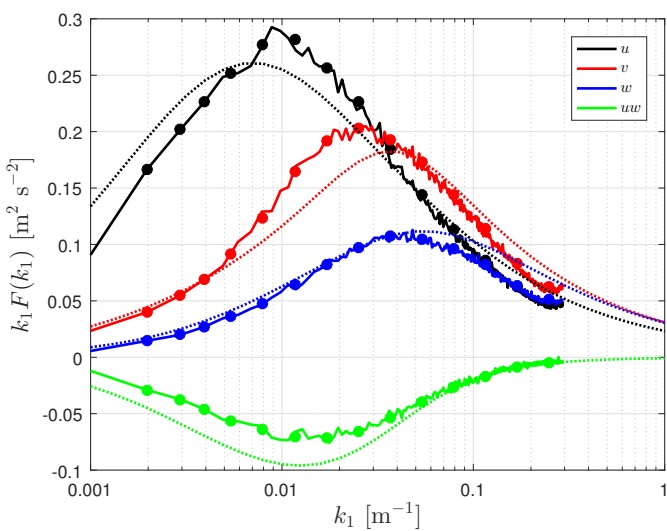

**Figure 9.** Power spectrum for different velocity components. The solid lines show the ensemble-average spectra of all 10-min sonic-anemometer spectra, the markers the $k$-based logarithmically-average spectra of all 10-min spectra, and the dotted lines a fit to the spectra using the Mann model

Due to the uncertainty on the sonic-derived statistics, we will use the cup-anemometer variance as a proxy for $\sigma_u^2$. However, we will use the sonic-based Mann parameters for comparison with the lidar-based Mann parameters (and for estimations of $\sigma_{v,w}^2$ and $\overline{u'w'}$) because it is the only reference we have for three-dimensional turbulence measurements.

### 5.3 Undersampling and noise removal

Although the variances of a velocity time series sampled over a 10-min period at a frequency $f_s$ of 0.2 or 0.06 Hz are not statistically different from those estimated from 1 or 10 Hz records, aliasing and noise might appear both in the sonic-anemometer and the lidar radial velocity spectra. Figure 10-left shows the Avent radial velocity spectrum that has been ensemble-averaged




from all the 10-min observed spectra for each of the beams. We conjecture that the increase in the spectral densities at high frequencies is due to noise. Figure 10-right shows the effect of a noise filter, which is based on the method by Kirchner (2005), on the ensemble-average Avent radial velocity spectra.

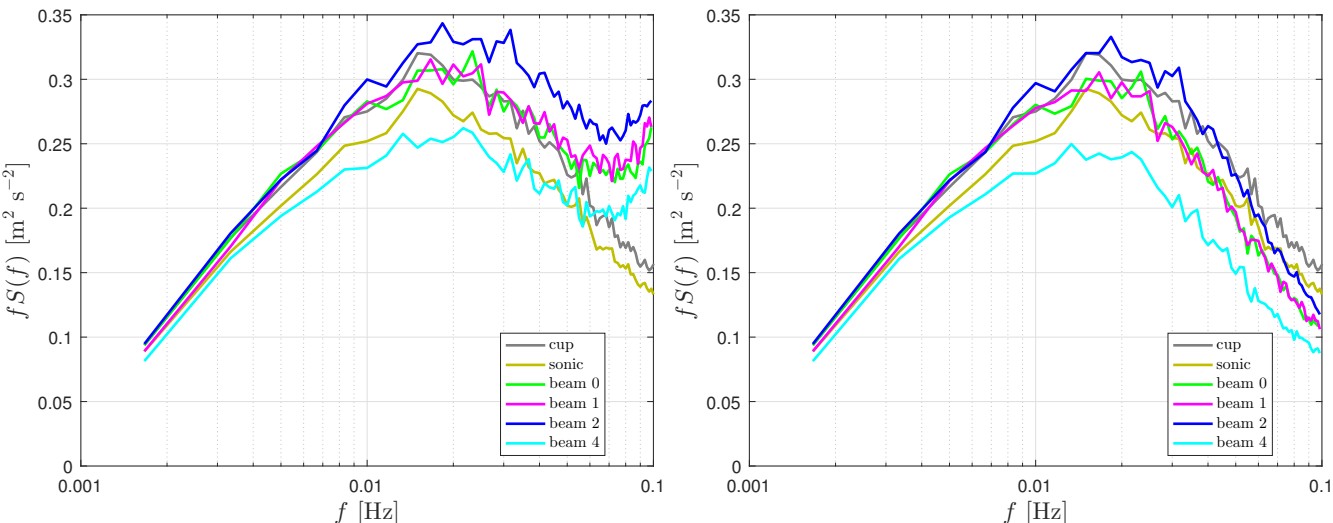

**Figure 10.** Ensemble-average spectrum of all 10-min Avent radial velocity spectra (per beam), sonic-anemometer $u$-spectrum and 80-m cup-anemometer spectrum. Original (left frame) and noise-filtered lidar radial velocity spectra (right panel)

The noise filter seems to recover the shape of the spectrum. However, when tested on the 18-s sonic ensemble-average $u$-spectrum (not shown), the filter highly distorts the shape and the peak of the spectrum. Therefore, we focus the spectra analysis on the measurements performed at $f_s \geq 0.2$ Hz, i.e. we exclude the ZephIR radial velocity spectra for the analysis.

Figure 10 also shows that for these ensemble-averages, the spectral density of beam 2 is the highest, followed by that of beams 0 and 1, and then that of beam 4. This behavior might be due to three reasons. Excessive rolling of the Avent, so that beam 2 points higher than beam 1, that the turbulence characteristics at the position of beam 2 are rather different than those at the position of beam 1, or that there is yaw misalignment so that beam 2 points closer to the direction of the mean wind compared to beam 1 (see Fig. 5-left). Both ZephIR and Avent have tilt and roll signals and for the 10-min samples here analyzed the maximum absolute 10-min mean tilt and roll are only $0.56°$ and $0.31°$, respectively. Also, the very flat terrain characteristics should not have such an impact on the ensemble-average spectrum of two beams that point at the same height, like beams 1 and 2 in this particular case. So the most plausible explanation is that beams 2 and 3 are more aligned with the mean wind than beams 1 and 4.

In Fig. 10-right we can see that the spectrum of beam 2 is slightly higher than that of the 80-m cup anemometer and higher than that of the sonic anemometer (up to $f \approx 0.4$ and 0.7 Hz, respectively). Such behavior is expected for low $z_R/L$ values (see Fig. 3-top left) or under lidar misalignment conditions (see Fig. 5). We can also see that the cup-anemometer spectrum is higher than that of the sonic anemometer, as expected from the variance results in Fig. 8-right.




### 5.4 Horizontal wind-speed reconstruction

For both lidars we need to reconstruct the horizontal wind speed to estimate a 'lidar-effective' velocity at the specific range of the lidars, which can later be used for spectral analysis and for filtered along-wind variance estimates. We use a simplified version of the linear-gradient model of Hardesty et al. (1981),

$$v_r(\theta) = -\cos\varphi \left( u + R_d \frac{du}{dz} \sin\theta \right) + v\cos\theta\sin\varphi, \tag{21}$$

where $R_d$ is the radius of the disc formed by the scanning pattern at the given range, to estimate $u$, $v$, and the vertical gradient of the along-wind component, $du/dz$. In Eqn. (21), we ignore $w$ and other vertical and horizontal gradients of the wind components because their contribution is small. For both lidars, the beams selected in Sect. 5.1 are used for the reconstruction, which can be done on the time-series basis or the 10-min averages. Figure 11 shows the results of the lidar-based reconstruction on all 10-min means compared to the 80-m cup anemometer and between the lidars; for both lidars we show the horizontal wind speed magnitude but when using the mean wind speed we obtain the same results. Same results, regarding the linear regression and $R^2$ (not shown), as those given in Fig. 11-left are found when comparing the radial velocity of beam 0 with the 80-m cup-anemometer wind speed on a 10-min basis.

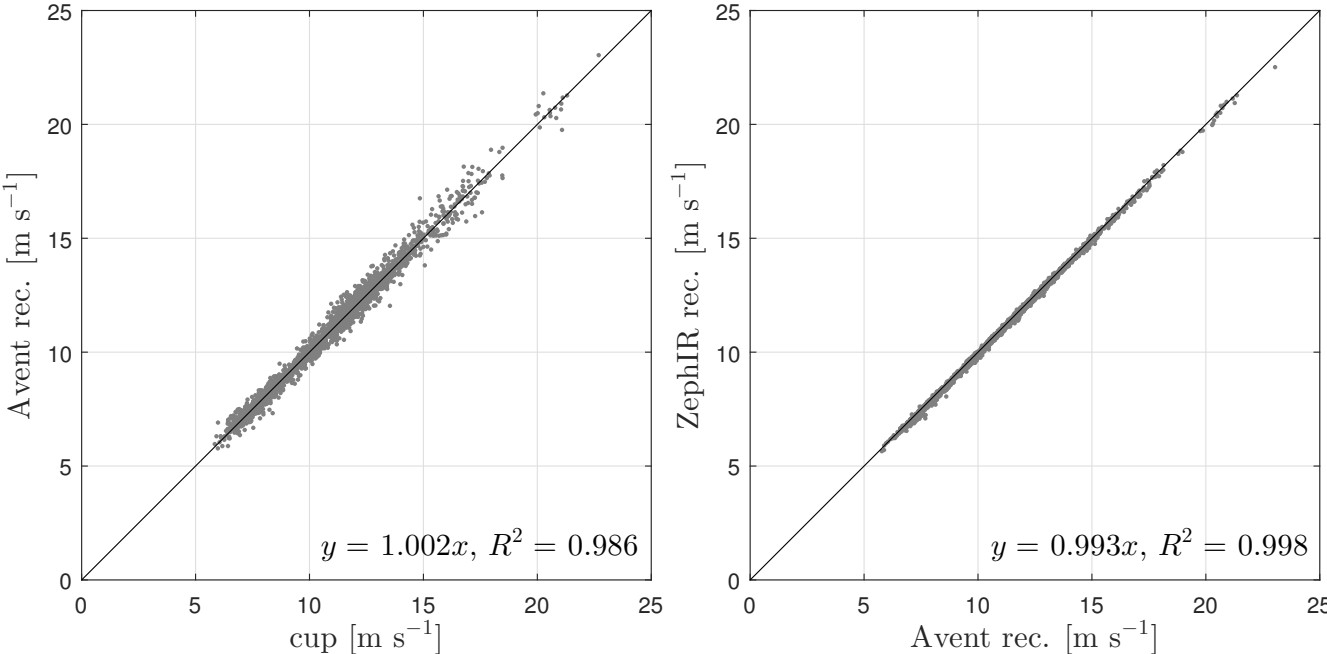

**Figure 11.** Comparison of reconstructed and 80-m cup anemometer horizontal wind speeds. (Left frame) cup anemometer against Avent. (Right frame) Avent against ZephIR. Each 10-min sample is shown in grey markers, a 1:1 line is shown for guidance in black, and the results of a linear regression through the origin and $R^2$ are also given





### 5.5 Ensemble-average Doppler radial velocity spectrum

The Doppler-spectrum analysis is performed over all the 2273 10-min periods using the ZephIR data (the Doppler spectrum information is not available for the Avent). While each of the 10-min radial velocity time series per bin position is thresholded and despiked (see Sect. 5.1), we extract the normalized Doppler radial velocity spectrum for each of the samples within that
10-min and bin position. We then sum all the normalized Doppler spectra within the 10-min period and the resulting Doppler spectrum is normalized to unit area before we estimate the variance in two ways: by computing the second moment from the spectrum and by fitting a normal distribution to the spectrum to extract its variance. Figure 12 illustrates examples of ensemble-average Doppler spectra for different 10-min periods for the positions of bins 0 and 31, where we intentionally show 10-min radial velocity distributions with high and low mean values, and high and low variances, including double-peak distributions
(there are only a few of them). These few distributions give us an idea of the variety of turbulence characteristics of the dataset. Distributions with high and low radial velocities generally show high and low variances, respectively, as expected. Particularly in the examples, there is a 10-min period with very low variance for both bin positions with clear larger radial velocities for bin 0 compared to those for bin 31, indicating very high wind shear, which is normally associated to atmospheric stable conditions. This is an early morning 10-min period in late October, in which the sonic-derived $L_O$-value is 1.82 m, corresponding to
extremely stable conditions.

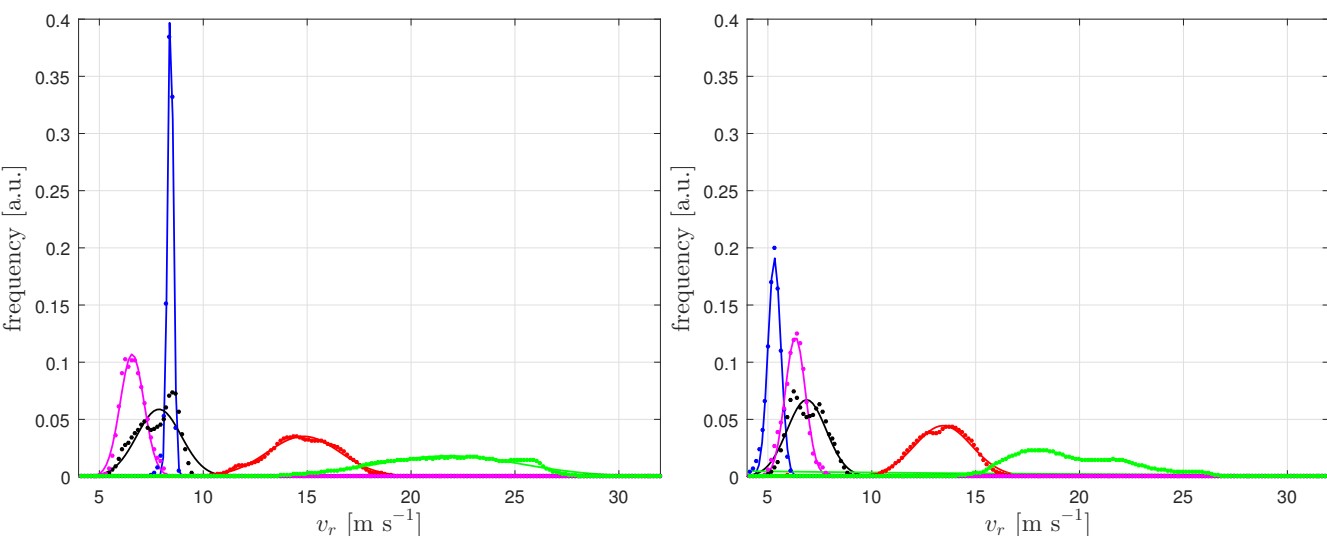

**Figure 12.** Examples of normalized Doppler radial velocity spectra measured over five 10-min periods with the ZephIR at the positions of bin 0 (left frame) and bin 31 (right frame). The markers show the observed distributions and the solid lines a normal fit



## 6   Results

The results are divided into five parts. In Sect. 6.1, we illustrate the main turbulence characteristics of the site, which we use to classify the data in a number of atmospheric-stability and wind-speed ranges. In Sect. 6.2, we intercompare the ZephIR estimates of variances and co-variances using the unfiltered lidar radial velocity variances with the cup- and sonic-anemometer estimates. Section 6.3 shows the effect of the noise-filter on the Avent radial velocity variance for the atmospheric-stability and wind-speed ranges. In Sect. 6.4, we explore the effect of atmospheric stability on both the sonic and the lidar radial turbulence spectra and intercompare the Mann parameters derived from both types of spectra. Finally, in Sect. 6.5, we perform the same exercise as in Sect. 6.4 but on the basis of the wind-speed ranges.

### 6.1   Turbulence characteristics

Figure 13 shows the overall turbulence characteristics of the site based on cup- and sonic-anemometer observations using the 2273 10-min concurrent data. In the left frame, we illustrate the behavior of the turbulence intensity, $\sigma_U/U$, with wind speed using the 80-m cup-anemometer measurements; wind speeds are in the range $\approx$5–23 m s$^{-1}$ with low $\sigma_U/U$ values within the low wind-speed range and $\sigma_U/U$ increasing with wind speed. In the right frame, we illustrate the behavior of the dimensionless wind shear, $\phi_m = (\kappa z/u_*)\,\partial U/\partial z$, with dimensionless atmospheric stability, $z/L_O$; we use the cup-anemometer wind-speed measurements at 78 and 56 m to estimate $\partial U/\partial z$ ($\approx \Delta U/\Delta z$) and the sonic-derived $u_*$- and $L_O$-values to compute $\phi_m$. Figure 13-right shows that the atmosphere during the analyzed period is mostly stable ($z/L_O > 0$) and that, as expected, $\phi_m$ increases with increasing $z/L_O$. Such atmospheric conditions explain the low $\sigma_U/U$ values for low wind speeds. In Fig. 13-left, we include a prediction of $\sigma_U/U$ using Eqn. (20) with $\sigma_U = 2.5u_*$ and $z_o = 0.012$ m, which fairly agrees with the data for high wind speeds only, as expected. In Fig. 13-right, we include, for comparison only, the prediction $\phi_m = 1 + 4.7z/L_O$ from surface-layer theory (Högström, 1988) that is offset with the data because $\phi_m$, and so $z/L_O$, are estimated at a mean height of $z = 67$ m with only two wind-speed observations 22-m apart, whereas the turbulence estimates are from the sonic anemometer at 76 m.

Based on the observed turbulence characteristics and knowing that we need to average a number of 10-min spectra to be able to robustly extract the Mann parameters (Peña et al., 2010), we classify the concurrent data into ten classes as illustrated in Table 1, ensuring that there are a close to 100 10-min samples per class as a minimum. From the atmospheric-stability classes, we can see that the data comprise mainly stable conditions with stability 1 being the only close-to-neutral class ($\langle 1/L_O \rangle^{-1} = 1072$ m). The more stable the atmospheric conditions the lower the wind speed and the friction velocity, as expected. Most of the data range within the stability 2 class ($\langle 1/L_O \rangle^{-1} = 450$ m), i.e. most of the observations are nearly stable. From the wind-speed classes, we observe most of the data within a high-speed range (11–13 m s$^{-1}$) and, similarly to the stability classification, the lower the wind speed the more stable the atmosphere and so the lower the friction velocity. Interestingly, for the speed 1 class, $\langle 1/L_O \rangle^{-1} = 8.15$ m, which is much lower than the average stability of the most stable class (stability 5).




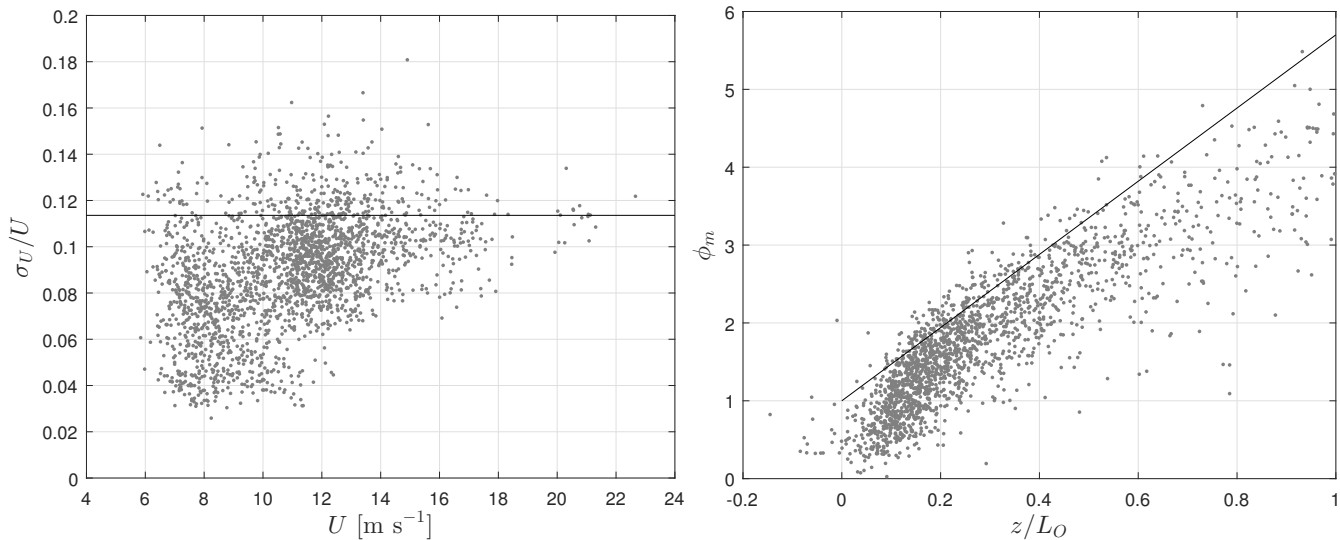

**Figure 13.** (Left) turbulence intensity $\sigma_U/U$ as function of mean wind speed $U$ from the 80-m cup-anemometer observations. (Right) dimensionless wind shear $\phi_m$ as function of dimensionless stability $z/L_O$ based on the sonic- and cup-anemometer observations. The grey markers show 2273 10-min concurrent samples and the solid lines are theoretical predictions (see text for details)

**Table 1.** Atmospheric-stability and wind-speed classes and ranges based on the cup- and sonic-anemometers' observations (see text for details). The ensemble-average values of the dimensionless stability, wind speed, and friction velocity per range are also provided. $z = 67$ m is here the mean height used for the dimensionless wind-shear and dimensionless atmospheric-stability estimates

| class | $z/L_0$ | no. of 10-min samples | $\langle z/L_O \rangle$ | $\langle U \rangle$ [m s$^{-1}$] | $\langle u_* \rangle$ [m s$^{-1}$] |
|---|---|---|---|---|---|
| stability 1 | −0.1–0.1 | 225 | 0.0625 | 12.75 | 0.68 |
| stability 2 | 0.1–0.2 | 629 | 0.1489 | 12.54 | 0.61 |
| stability 3 | 0.2–0.3 | 350 | 0.2435 | 11.34 | 0.48 |
| stability 4 | 0.3–0.4 | 225 | 0.3475 | 10.71 | 0.42 |
| stability 5 | 0.4–0.5 | 153 | 0.4457 | 10.02 | 0.35 |
| class | $U$ [m s$^{-1}$] | no. of 10-min samples | $\langle U \rangle$ [m s$^{-1}$] | $\langle z/L_O \rangle$ | $\langle u_* \rangle$ [m s$^{-1}$] |
| speed 1 | 5–7 | 93 | 6.65 | 8.2174 | 0.21 |
| speed 2 | 7–9 | 516 | 7.98 | 4.2195 | 0.23 |
| speed 3 | 9–11 | 506 | 10.07 | 0.8401 | 0.37 |
| speed 4 | 11–13 | 741 | 11.94 | 0.3015 | 0.52 |
| speed 5 | 13–15 | 278 | 13.82 | 0.1593 | 0.64 |




### 6.2 Unfiltered lidar turbulence

Based on the ZephIR configuration ($\varphi = 15.05°$), we are able to predict all variances' ratios $\sigma_{v_r}^2/\sigma_{u,v,w}^2$ using the Mann model with a given $\Gamma$-parameter for the unfiltered lidar radial velocity variances, i.e. using Eqn. (14) with $z_R/L = 0$. This is a procedure similar to the one we use for the results in Fig. 4-right. Figure 14 shows a comparison of the ZephIR 'unfiltered'

5   radial velocity variances (for bins 0 and 31) with the cup-anemometer variances for all the 2273 10-min data, together with the Mann-model prediction using $\Gamma = 3$. We present variance estimations that are computed from the normal distribution fit to the average normalized Doppler spectrum, instead of those calculating the second moment from the spectrum, since the latter method is more sensitive to 'spurious' data that appear far from the area where most radial velocities are concentrated. This is particularly seen for the lower bins (18 and 31) and might be due to non-filtered blade-obstructed data, noise, or sudden jumps

10   in the radial velocity within the 10-min period.

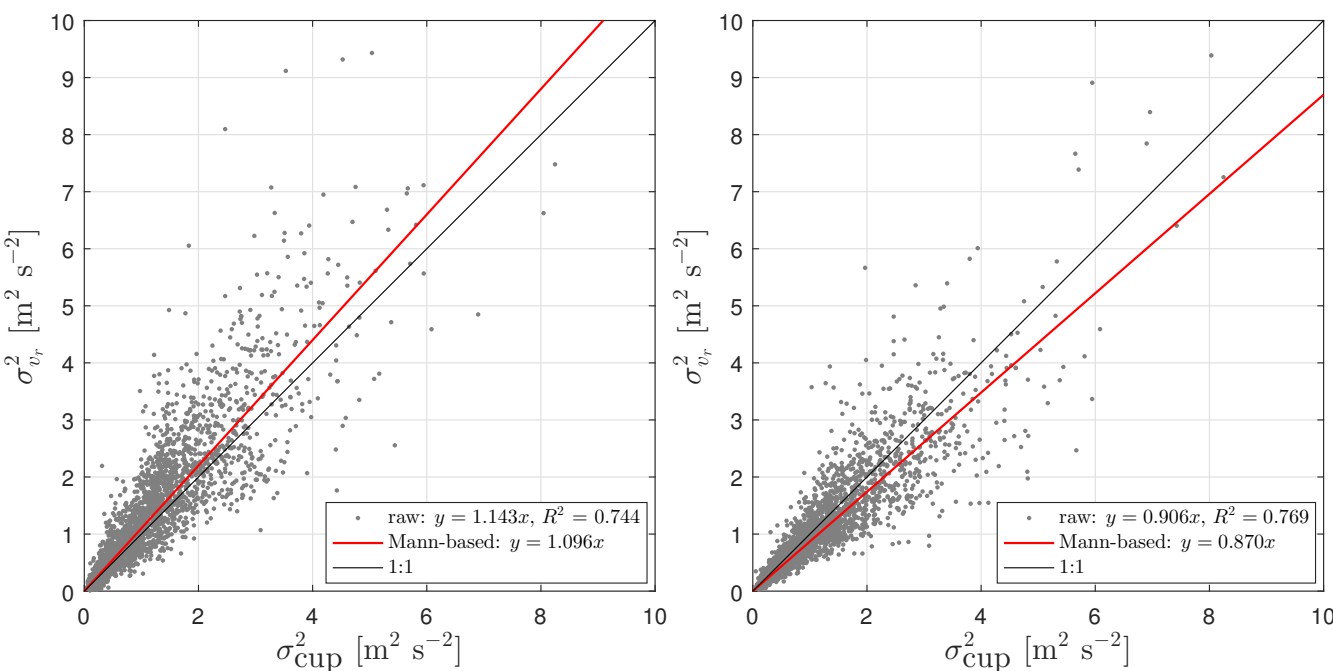

**Figure 14.** Comparison of the 80-m cup-anemometer and the unfiltered ZephIR radial velocity variances for bins 0 (left frame) and 31 (right frame). We show a 1:1 line for guidance and the predictions of the Mann model using $\Gamma = 3$. Results of a linear regression through the origin and $R^2$ are also given.

As expected, based on the results in Fig. 4, the top (bin 0) and a lower beam (bin 31) show a higher and lower variance, respectively, than that of the '$u$'-component (in quotation marks because we use the cup-anemometer measurements). The Mann-model-based results slightly underpredict the ratio $\sigma_{v_r}^2/\sigma_u^2$ for these two beams. Reducing the value of $\Gamma$ or accounting for misalignment improves the predictions; e.g. with $\Gamma = 2.5$ and $\beta = 0°$ the Mann-model results predict biases of 11% and





−13% for bins 0 and 31, respectively (not shown). It is important to highlight that the original (filtered) radial velocity variances for these two bins are 13% and 31% lower than the cup-anemometer measurements (not shown) with slightly higher $R^2$-values, 0.785 and 0.798, respectively.

Further, we can also estimate $\sigma_{u,v,w}$ and $\overline{u'w'}$ for each 10-min period through a least-squares fit of Eqn. (17), that does not depend on the Mann parameters but assume homogeneous turbulence within the scanned volume, using the unfiltered radial velocity variances. Figure 15-left shows the estimate of $\sigma_u^2$ based on the unfiltered radial velocity variances of all bins without accounting for misalignment compared to $\sigma_{\text{cup}}^2$. The lidar-variance estimate is only 2% larger than the cup-anemometer value and the $R^2$-value is higher than that of any other comparison between cup-anemometer and lidar beam radial velocity variances (filtered or not).

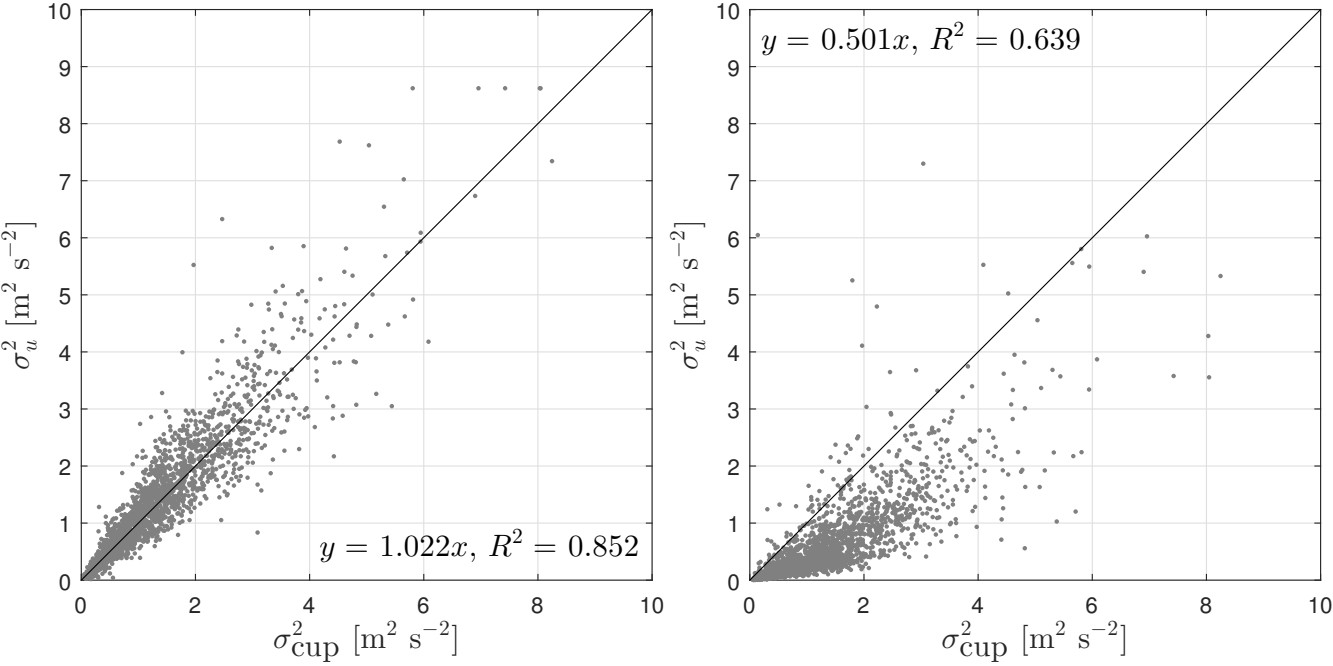

**Figure 15.** Comparison of the 80-m cup anemometer and the unfiltered (left frame) and filtered (right frame) $u$-variances from the ZephIR estimated under the assumption of homogeneous turbulence within the measurement volume (see text for details). We show a 1:1 line for guidance. Results of a linear regression through the origin and $R^2$ are also given

In Fig. 15-right we show a similar comparison to the plot in the left panel but for the 'filtered' $u$-variance, which was computed by reconstructing the $u$- and $v$-components, as described in Sect 5.4, using the ZephIR measurements on the seven bins, but from the 18-s radial velocity measurements. The comparison with the filtered values shows poor agreement with a 50% underestimation of the variance by the ZephIR. However, reconstructed $u$-velocities from the 18-s radial velocities and averaged within 10-min periods compare well with the reconstructed values from the 10-min means; the mean bias is 0% and $R^2 = 0.999$ (not shown).





We also compare the lidar-derived $\sigma_{v,w}$ and $\overline{u'w'}$-values with the sonic-anemometer estimates; the biases are very high and $R^2$-values are very low (not shown). This is not surprising given the weight of the $\sigma_{v,w}$ and $\overline{u'w'}$-terms in Eqn. (17) when using low $\varphi$-values. With this lidar configuration, the reconstruction of the $v$-component is not sound either, e.g. from Eqn. (21); the yaw misalignment based on both the Avent and ZephIR reconstructed $u$- and $v$-components shows poor agreement when

compared to the difference between the wind-vane and the turbine-yaw 10-min signals.

We can also estimate $\sigma_u^2$ through a least-squares fit of Eqn. (17) but using the unfiltered radial velocity variances of the horizontal bins (12 and 37) only and the comparison with $\sigma_{\mathrm{cup}}^2$ shows similar results (bias of 3% and $R^2 = 0.842$). This indicates, firstly, the small but positive effect of adding the top and lower beams variances and, secondly, that the contributions of other velocity components are not that significant for the estimation of $\sigma_u^2$ with the actual lidar configuration. Accounting

for misalignment does not improve the variance comparison (the bias increases from 2% to 7%).

### 6.3   Effect of the noise filter on the lidar variances

We also classify the 10-min 80-m cup anemometer variances and Avent radial velocity spectra into the classes given in Table 1, ensemble-average the spectra within each class, and compute the variance of each ensemble-average spectrum. The comparison of such variances, for each Avent beam, is illustrated in Fig. 16 (raw). We also show a similar comparison but for the noise-

filtered Avent radial velocity ensemble-average spectra. Further, we include the prediction $\sigma_{v_r}^2/\sigma_u^2$ based on the Avent lidar configuration using the Mann model with fixed Mann parameters (same as those found in Sect. 5.2 using the ensemble-average sonic-anemometer velocity spectra).

When the noise-filter is applied, the ratio $\sigma_{v_r}^2/\sigma_{\mathrm{cup}}^2$ is well predicted by the Mann model. The largest difference is observed for beam 4 but this is because the noise-filter highly reduces the variance for one particular class only. For beams 1 and 2, the

Mann model predicts the same $\sigma_{v_r}^2/\sigma_{\mathrm{cup}}^2$ value as here we do not take into account lidar misalignment.

### 6.4   Effect of atmospheric stability on turbulence spectra

The ensemble-average sonic and Avent radial velocity spectra are used separately to extract two independent sets of Mann parameters for each of the atmospheric stability classes in Table 1 by fitting the sonic- and lidar-based LUTs computed through the use of Eqns. (6) and (14). Figure 17 shows the results of the two stability classes most far apart (stabilities 1 and 5).

For the stability 1 class, the Mann model agrees well with the sonic velocity spectra and for stability 5 the differences between the model and the sonic-anemometer observations are larger, as expected, since the Mann model was developed for near-neutral atmospheric conditions. Both the sonic observations and the Mann model show the spectral peaks to move to higher wave numbers with increasing stability because the size of the turbulence eddies decreases with stability in agreement with the study of Peña et al. (2010). The lidar radial velocity spectra also show similar features as the sonic-based spectra;

higher normalized spectral densities for the most stable compared to the close to neutral class and spectral peaks that move to higher wave numbers with increasing stability. The former feature might be due to the way we normalize the spectra: we use of the 80-m $u_*$ value instead of one close to the ground where surface-layer scaling is more valid, particularly for stable







**Figure 16.** Comparison of the 80-m cup anemometer with the Avent radial velocity variances for different beams for the ten turbulence classes (filled circles) in Table 1. Raw and noise-filtered data are shown as well as a 1:1 line (for guidance) and the prediction of the Avent filtered radial velocity variance based on the Mann model using $\Gamma = 3.00$, $L = 35.38$ m, and $\beta = 0°$





**Figure 17.** Normalized power spectra of the different velocity components based on the sonic-anemometer observations (left frames) and of the Avent radial velocity for different beams (right frames). The top panels show the results for the first stability range (stability 1) and the bottom panels for the last stability range (stability 5)





conditions. The agreement of the Mann-model based spectra also deteriorates with stability but the lidar-based LUT seems to follow fairly well the behavior of the radial velocity spectra for these two classes.

In Fig. 18, we show the results of the Mann parameters extracted from the ensemble-average sonic and lidar radial velocity spectra for all atmospheric stability classes. $\Gamma$ slightly decreases with stability (based on the sonic-anemometer data) and the
lidar-based value closely follows the sonic-based one, with best agreement at the highest stability range. The sonic-based $\alpha\varepsilon^{2/3}$-parameter slightly decreases with stability and, for the near-neutral stability class, the lidar-based value is close to the sonic-based one. A similar feature is found for the $L$-parameter; both types of data show a very close value for near-neutral conditions and the sonic-based value slightly decreases with stability as expected. The increasing differences between the sonic- and the lidar-based $\alpha\varepsilon^{2/3}$- and $L$-parameters with stability are interconnected. We cannot expect to measure eddies
below the size of the lidar probe volume, which in this case means that we are not able to accurately estimate the length scale when $L \lessgtr z_R$. This occurs already at the stability 3 class. And these two Mann parameters are, in practical terms, scaling factors in the velocity spectra as seen from Eqn. (7), and so an underestimation of $L$ generally leads to an overestimation of $\alpha\varepsilon^{2/3}$ when fitting the lidar-based LUT.

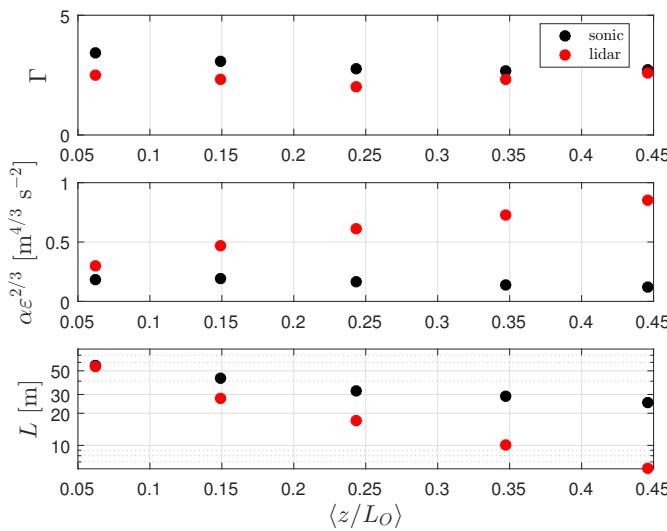

**Figure 18.** Mann parameters for a number of atmospheric-stability conditions (see Table 1) derived from sonic anemometer and lidar radial velocity spectra

We also have to notice that when using this type of lidar configuration, we are extracting turbulence information from the
radial velocity spectra of beams, whose spectral densities are rather close (since all beams measure a close to $u$-spectrum), whereas in the case of the sonic-anemometer observations we use three auto-spectra and a cross-spectrum that are relatively far apart in terms of spectral densities. This issue is discussed further in Sect. 7.





## 6.5 Effect of wind speed on turbulence spectra

We now perform a similar procedure as that in Sect. 6.4 but for each of the wind-speed classes in Table 1, and the results of the two wind-speed classes most far apart (speeds 1 and 5) are shown in Fig. 19. For the speed 1 class, the Mann model does not agree with the sonic velocity spectra as well as it does when compared to the speed 5 class, as expected, since the
atmospheric conditions are closer to neutral for the latter class. Both the sonic-anemometer observations and the Mann model show spectral peaks that move to lower wave numbers with increasing wind speed because of the combined effect of stability and wind speed; the larger the turbulent eddies the higher the wind speed and the lower the stability.

The lidar radial velocity spectra also show similar features as the sonic-based spectra; lower normalized spectral densities for the high wind compared to the low wind class and spectral peaks that move to lower wave numbers with increasing wind
speed. The agreement of the Mann model-based spectra deteriorates with decreasing wind speed but the lidar-based LUT also seems to follow fairly well the behavior of the radial velocity spectra for these two classes (similarly as it does when comparing spectra for the range of stability classes).

In Fig. 20, we show the results of the Mann parameters but for the wind-speed classes. $\Gamma$, based on the sonic-anemometer data, is rather constant with wind speed, a behavior already observed by Peña et al. (2010) for the same height and the lidar-
based value agrees well with the sonic-based one for all wind-speed classes, particularly the two low wind-speed classes. Similar to the results from the atmospheric-stability classes, the differences between the sonic- and the lidar-based $\alpha\varepsilon^{2/3}$- and $L$-parameters are larger than those for $\Gamma$, but for these wind-speed classes the $\alpha\varepsilon^{2/3}$-parameter does not differ largely under the classes where $L$ differs the most, i.e. speed classes 1 and 2, where the average conditions are very stable. The turbulence characteristics under these two classes are similar and $L$ is higher within the speed 1 compared to the speed 2 class. The highest
differences in the estimations of $L$ are also found for those classes in which $L \lessgtr z_R$ (speed classes 1–3).

## 7 Discussion

It is important to notice that some of the differences between turbulence statistics estimated from the sonic-, cup-anemometer, and lidars' measurements are not only due to the way they probe the atmosphere but also because the lidar measurements are affected by optical and instrumental noise (and by the blades, hard targets, and fog among others), the cup- and sonic-
anemometers are inherently affected by flow distortion from the mast structure and the instrument itself, which we do not take into account, and that there are differences in the heights of the measurements. For example, the axes of the lidars pointed close to hub height when the wind turbine was operating, and the 80-m cup and sonic anemometer are 1.8 and 5.8 m below hub height, respectively. Also, the mast is 111 m from the range that we use to extract the lidar measurements, when the wind is directly from the mast to the turbine. Wind speeds, variances, and velocity spectra from the mast and the lidars' selected range
are expected to be comparable, due to the topographic conditions of the site for the selected wind directions, but not equal.

Further, we assume turbulence to be homogeneous within the lidar scanning area, both when extracting the Mann parameters and when studying the unfiltered turbulence. This is a rather simplistic assumption as shown in the study by Peña et al. (2010), in which the Mann parameters are extracted from sonic-anemometer measurements at different heights. However, we expect





**Figure 19.** Similar to Fig. 17 but here the top panels show the results for the first wind-speed range (speed 1) and the bottom panels for the last wind-speed range (speed 5)



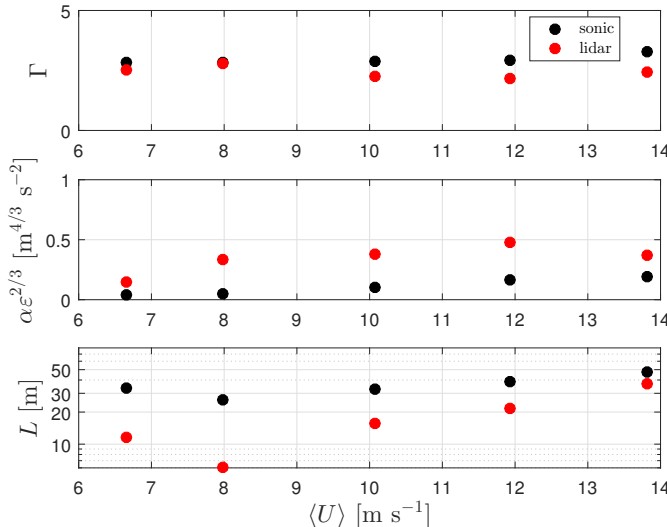

**Figure 20.** Mann parameters for a number of wind-speed ranges (see Table 1) derived from sonic anemometer and lidar radial velocity spectra

that such an assumption results in turbulence parameters that are more representative for the turbine operation as they are estimated from measurements over a larger area.

In Sects. 6.4 and 6.5, we show normalized power spectra for the two most 'extreme' classes in order to understand the spectra behavior for the changing atmospheric and wind-speed conditions; spectra results for the other classes are not shown but lie in between, as illustrated from the derived Mann parameters in Figs. 18 and 20. In Sect. 6.4, we mention that part of the problem of extracting the Mann parameters from the current lidar measurements is the small difference between the beams' radial velocity spectra, all being relatively close to the $u$-spectrum. The Mann model needs more than one-component spectra to fit the LUT to measurements/simulations, otherwise the Mann parameters are ill-determined.

We find very good agreement between the along-wind variance estimate of the ZephIR (when using the ensemble-average Doppler radial velocity spectrum) and the cup-anemometer measurement but for the other velocity-component variances and co-variances the biases are too large. But, can we improve such estimates, e.g. increasing the cone angle $\varphi$? On one hand, one can make the theoretical exercise of predicting $\sigma^2_{u,v,w}$ and $\overline{u'w'}$ from the Mann model (with a given set of Mann parameters). In parallel, we can use Eqn. (14) with $z_R = 0$ to estimate the unfiltered $\sigma^2_{v_r}$ for different beams and use Eqn. (17) to estimate $\sigma^2_{u,v,w}$ and $\overline{u'w'}$ from the unfiltered beam variances. If we compare the former predicted with the latter estimated variances, e.g. using a four-beam lidar ($\theta = 0$, 90, 180 and 270°) with $\varphi = 15°$, the result for the $u$-variance is a 2% bias, whereas the $v$- and $w$-variances show biases larger than 50%. The result for the $v$- and $w$-components improves when increasing $\varphi$; the biases for both components' variances are below 20% for $\varphi = 60°$ but the bias deteriorates for the $u$-variance with increasing $\varphi$. If a central beam is added and we are able to extract the unfiltered variance of this beam, i.e. $\sigma^2_u$, the comparisons are unbiased for all velocity components (no matter the value of $\varphi$).




On the other hand, using a lidar with $\varphi = 60°$ increases the relative differences between the radial velocity spectra densities of the beams, e.g. with the current Avent configuration as it uses a central beam. Figure 21-left shows that with such cone angle, the central beam spectrum peaks close to the $u$-spectrum peak, and the lower beams peak at $\approx 20\%$ of the $u$-spectrum peak (with $\varphi = 15°$ the lower beams peak at $\approx 75\%$ of that of the central beam spectrum). This is mainly due to the large negative
contribution of $\Phi_{uw}$ for the lower beams, as shown in Fig. 21-right. The difference between the $u$- and $w$-spectra is $\approx 60\%$ only.

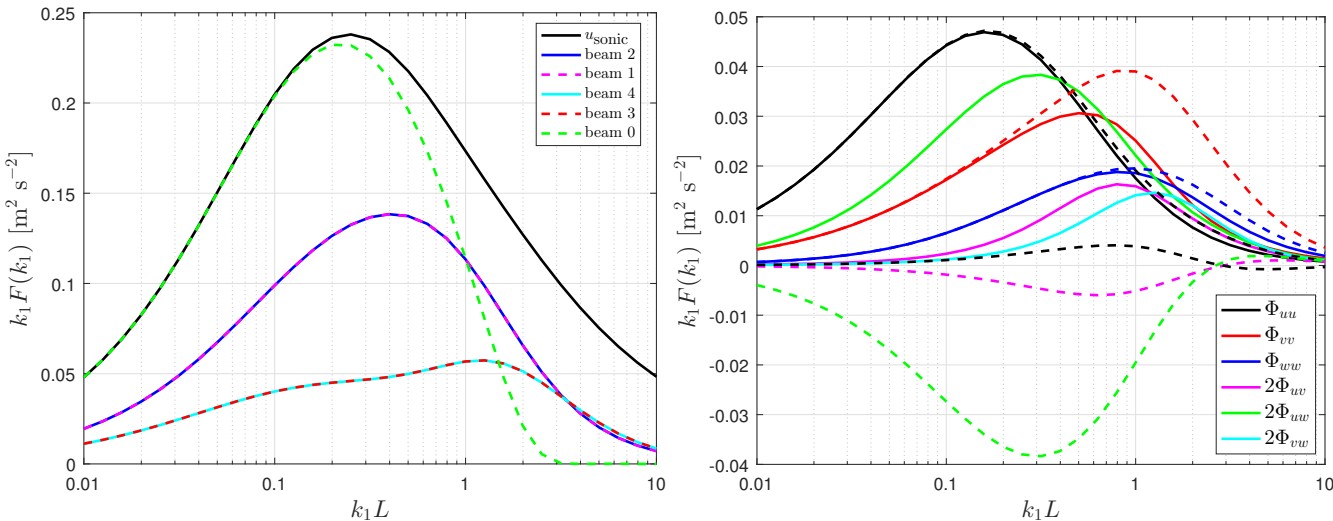

**Figure 21.** (Left panel) pulsed lidar radial velocity spectra for different beams. (Right panel) contributions of the spectral velocity tensor components to the lidar radial velocity spectrum for beams 2 (solid lines) and 3 (dashed lines). Values of $\beta = 0°$, $\varphi = 60°$, $\Gamma = 3$, $\alpha\varepsilon^{2/3} = 0.1\ \mathrm{m^{4/3}\ s^{-2}}$ and $z_R/L = 0.5$ are used for the computation

It is also important to highlight that wind turbine loads are directly impacted by turbulence, in particular $\sigma_u^2$. The Mann parameters add value for understanding the behavior of loads but are not critical (Dimitrov et al., 2017). In this study we demonstrate that $\sigma_u^2$ can be estimated by FL nacelle lidars, and current research demonstrates that lidar-based $\sigma_u^2$ values reduce
the gap between loads measurements and simulations.

## 8   Conclusions

We characterize turbulence using measurements from two types of forward-looking nacelle lidars that were mounted on the nacelle of a wind turbine. We compare such characteristics with those from sonic- and cup-anemometer measurements on a mast, which is 111 m from the lidar measurement range when the turbine and mast are aligned with the wind (thus this distance
increases for other wind directions). By using information of the 10-min ensemble-average Doppler radial velocity spectrum, we are able to estimate 10-min unfiltered radial velocity variances of the beams of a CW lidar. These unfiltered beam variances





are well predicted by the Mann model. Assuming homogeneous turbulence within the lidar scanned area, $\sigma_{u,v,w}$ and $\overline{u'w'}$ are estimated from the unfiltered beam variances; comparison with the 10-min cup-anemometer variances reveals a 2% bias for the $u$-variance, whereas the biases are very high for the other velocity components.

We divide the 10-min time series and the sonic-anemometer and lidar beam radial velocity spectra into atmospheric-stability and wind-speed classes based on the mast measurements. Most of the conditions are stable and relatively windy. We observe that the pulsed lidar beam variances are affected by noise as clearly seen in the lidar radial velocity spectra. Therefore, we noise-filter the lidar beam spectra and the resulting variances show very good agreement with the prediction using the Mann and spatial averaging model.

We also extract the Mann parameters from sonic-anemometer and lidar beam radial velocity spectra and intercompare them for each of the classes. Under high wind and near-neutral atmospheric conditions the agreement is good, and the differences increase with higher stability and lower wind speed, where the Mann model also has limitations fitting the sonic-anemometer velocity spectra. This is partly because increasing stability and decreasing wind speed results in turbulence length scales comparable to or lower than the length of the lidar probe volume. We suggest to improve lidar-based Mann-parameter estimations by increasing the lidars' cone angle, keeping always a central beam, which will also aid to the estimations of the non wind-aligned velocity variances and covariances.

*Acknowledgements.* Funding from Innovation Fund Denmark, grant number 1205-00024B, to the UniTTe project (www.unitte.dk) is acknowledged. We would also like to thank for the continuous technical support by Antoine Borraccino and Andrea Vignaroli, DTU Wind Energy.



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
