# Peer review of "Turbulence characterization from a forward-looking nacelle lidar"

_Wind Energy Science, 2016_

## Referee Comment (RC1) · W. Bierbooms (Referee) · 18 Jan 2017

This is an excellent paper on measuring turbulence with a lidar. I have just a few remarks in order to improve the readability.

The division of section 2 in just one subsection, 2.1, is rather odd and can be omitted.

The direction of the lines in Figure 1 should be clearly indicated, e.g. by adding a plane and/or a front view. The different colours should be mentioned in the caption.

Page 6, line 4: "This is expected .." It can be explained in more detail why this is.

Page 9, Figure 4: in order to avoid confusion with figure 4a other colours should be used in figure 4b; furthermore $z_R/L=0$ should be added (in the caption)

Page 13, line 15: explain the figure-of-eight

[Figure]

Page 14, mention in the caption of Figure 7 (left) why beam 3 is omitted.

Page 18, first sentence: explain "lidar-effective velocity"

Page 19: explain (e.g. in an appendix) the normalized Doppler radial velocity spectrum. Furthermore, it is unclear to me how the variance can be estimated after normalisation (line 6, page 19)

Page 22, last complete sentence: it is not clear to me why the ratio is underpredicted; in Fig. 4a I notice a ratio of about 0.82 (red dashed line); in fig 14 b a slope of 0.87 is indicated in the fitted red line (Mann-based), so an overprediction.

Typo's etc. Caption Figure 4: Change "divided by the lidar..." into "divided by the variance of the lidar ..."

Page 17, line 11: the reference to Fig 5-left is wrong

Page 17, line 17; the frequencies should be 0.04 and 0.07 Hz.

Page 20, line 15: change into:"... and the sonic-derived u_star to compute L_O and phi_m values"

Page 24, line 31/32 change into: "we make use of"

Page 31, figure 21 right: probably a wrong colour is used for PHI_vw; it should be light blue (instead of black)

There is something peculiar with figure 14: in the left figure 2 data points are visible in the column for sig^2_cup between 6 and 8 m^2/s^2; and 2 data points in the column between 8 and 10. In the right figure it are 4 and 2 data points resp. (In Fig 15 left it are resp 4 and 3 data points and in 15 right 4 and 3.)

---

## Referee Comment (RC2) · Anonymous Referee #2 · 24 Jan 2017

The characterization of 3-D turbulence from lidar for more accurate quantification of the rotor-disk wind resource and for predicting wind power generation and loads is in critical need, especially as lidar become more prevalent across wind farms. The paper by Pena et al. presents two methodologies for improving our understanding of and measurement techniques for obtaining accurate estimates of atmospheric turbulence across a turbine rotor-disk using forward-looking nacelle-mounted lidar systems. Lidar measurements of the radial-velocity, as well as the velocity spectra, and derived variances, are compared against tower-mounted cup anemometer and sonic anemometer measurements. Results are presented for both pulsed and continuous wave lidar systems and as a function of beam orientation, atmospheric stability, wind speed class, and cone angle. The authors conclude from the results that the use of a central beam and a larger cone angle would improve the accuracy of lidar turbulence measurements.

This topic is of high interest to the wind energy community as research investigating the sizes of turbulence and effects of 3-D turbulence on power generation and fatigue loads is currently being presented by numerous research groups, often offering alternative methodologies for obtaining accurate estimates of turbulence from lidar. I recommend acceptance of the manuscript after revisions, largely to help with the clarity of the results findings.

Major points: The manuscript is currently very long; results are not presented until page 20. I recommend that the authors consider whether the material presented in Section 2: General Background can be shortened. Does this information exist in earlier publications and can be largely cited here instead of explained in detail? The same comment is relevant for Section 3. These sections would be easier for the reader to digest if they were made more concise.

The use of sonic anemometry and cup anemometry is confusing throughout the results section. Please state that the sonic anemometer is 3-D. Only someone familiar with the CSAT3 would be aware of this since it is not mentioned in the text (as far as I can see). I recommend that a discussion is added either to Section 4 or to the Discussion Section which states the measurement differences between all of the instruments. This is briefly mentioned at the end of the manuscript but the point is important. A cup anemometer does not measure the three velocity components; instead mean horizontal velocity and variance is measured. Because of this, the reader is left wondering why the authors rely so heavily on the cup anemometer measurements for comparison to the lidar estimates of variance. At the very least a discussion needs to be included which outlines the limitation of deriving turbulence measurements from a cup anemometer.

The discussion would be strengthened by comparing these results to prior studies that have derived or utilized methodologies for estimating turbulence from lidar. Many of these studies use vertical-profiling or scanning lidar, however they are still relevant. Examples include recent work by J. Newman et al. (2016) whose group examined

the accuracy of lidar variance against 3-D sonic anemometry. Lastly, please add additional discussion concerning the motivation for needing better lidar turbulence measurements. This is briefly addressed in the abstract "....useful to predict the loads on a turbine", however the connection between the two is left up to the reader without additional information. Also, please discuss the connection between turbulence and power generation as many recent studies have been published in this area.

Minor points: 1. Figure 1. Please connect the ends of the beam lines. It took this reviewer a long time to realize that beam #2 and beam #4 were not in the same plane. Also, why are 13 beams drawn for the CW system? Is this an arbitrary number?

2. P 5 Line 20. Please be more specific. Which frequencies does the lidar average out? This includes "most eddies" of what size?

3. P 6 Line 3. Elaborate on why the sonic u-spectrum is considered "ideal".

4. P 6 Line 6-7. "...and upward pointing beam spectra is smaller than the differences between ordinary velocity-component spectra..." What are ordinary velocity-component spectra?

5. Page 6 and throughout, Please comment on what $Z_r/L$ represents. If I understand correctly $Z_r$ is constant for a particular lidar, so this ratio is a function of atmospheric stability?

6. Page 6 L 31. Aren't the two stress fluxes small because of the applied coordinate rotation?

7. Page 12 L 12-13. The last part of this sentence "...., five days were used to measure with the ZephIR..." is confusing and does not warrant inclusion. Better to leave this info out.

8. Page 13 L 30. The 5-s and 18-s time period were chosen to mimic the lidar sampling frequencies, right?

9 Figure 7, top right. What are the dots with a value of 1 along the y-axis? Are they blade interference? Better to remove these points from the figure and use the same y-axis for the left and right panels.

10. Page 15 Line 5. Please comment on the appropriateness of the logarithmic profile for the site here.

11. Figure 8. Left panel. Is this not showing evidence of overspeeding by the cup anemometer?

12. Page 15 L5-11. This section needs to be discussed in terms of instrument measurement technique differences and common errors associated with each measurement device. Is it not surprising that the cup anemometer is measuring higher variance than the sonic since it may be contaminated by the w component. It needs to be clear that horizontal variance from a sonic and cup anemometer are not expected to be equal, there will be a bias. Is this total horizontal or u?

13. Page 16 L 5. The reader may not know which scales are included in the inertial subrange. Please state.

14. Page 15 L 13. Doesn't local isotropy assume neutral conditions? If the conditions are stable, wouldn't this explain why the w spectrum is also lower than the u spectrum?

15. Figure 10 and text below. The sonic and cup anemometers appear to not suffer from noise at the high frequency end of the spectra. So why was the same noise filter applied to these data, especially since it "distorted the shape" of the sonic u-spectrum?

16. Figure 12. Please list the date periods that these lines correspond to. Also the stability conditions for each.

17. Page 20 L 25-30. Instead of using (1/L)-1, why not discuss z/L here since these values are in the table. Plus, isn't (1/L)-1 just L?

18. Figure 13 Left panel. Why is it assumed that sigma_u/U is constant as a function

[Figure]

of wind speed?

19 Table 1. I don't think mean values of z/L are meaningful, especially since the mean values you list are well above all of your stability classes. Try using median values instead.

20. Page 28 Line 22. Please talk about the fundamental differences in way the instruments measure velocity and turbulence. It is more than "due to the way they probe the atmosphere".

21. Page 30 Line 10. How do you get uw and w variance from a cup anemometer?

22. Page 32 Line 14. The authors conclude that a larger cone angle would improve estimates of turbulence, but doesn't this make the assumptions about flow homogeneity across the scanning cone less valid?

---

## Author Comment (AC1) · 2 Feb 2017

Dear Dr. Bierbooms,

Thank you very much for your opinion on the manuscript and for your detailed comments that are very helpful and appreciated.
Herewith our response to your comments in red: XXX--- the response ---XXX
* * *
This is an excellent paper on measuring turbulence with a lidar. I have just a few remarks in order to improve the readability.

The division of section 2 in just one subsection, 2.1, is rather odd and can be omitted.

XXX--- We agree with you. Now there is no subdivision of the section and has been shortened ---XXX

The direction of the lines in Figure 1 should be clearly indicated, e.g. by adding a plane and/or a front view. The different colours should be mentioned in the caption.

XXX--- The figure is now changed as suggested and the caption contains now details on the colors ---XXX

Page 6, line 4: "This is expected .." It can be explained in more detail why this is.

XXX--- We now provide a further explanation of this and also give an additional reference ---XXX

Page 9, Figure 4: in order to avoid confusion with figure 4a other colours should be used in figure 4b; furthermore $z\_R/L=0$ should be added (in the caption)

XXX--- Figure is changed as suggested and the suggested text is added to the caption ---XXX

Page 13, line 15: explain the figure-of-eight

XXX--- We expanded the explanation regarding the figure-of-eight ---XXX

Page 14, mention in the caption of Figure 7 (left) why beam 3 is omitted.

XXX--- Text is now added to the caption as suggested ---XXX

Page 18, first sentence: explain "lidar-effective velocity"

XXX--- The text was indeed unnecessary and so it is now removed ---XXX

Page 19: explain (e.g. in an appendix) the normalized Doppler radial velocity spectrum. Furthermore, it is unclear to me how the variance can be estimated after normalisation (line 6, page 19)

XXX--- Due to the length of the paper we do not want to include appendices, so we now made references to papers where these details can be found ---XXX

Page 22, last complete sentence: it is not clear to me why the ratio is underpredicted; in Fig. 4a I notice a ratio of about 0.82 (red dashed line); in fig 14 b a slope of 0.87 is indicated in the fitted red line (Mann-based), so an overprediction.

XXX--- Now we add "compared to the raw data" to that sentence because the "underprediction" is related to that. In Fig. 4a the ratio 0.82 is found for a theoretical bottom beam. Beam 31 is not exactly at the bottom so that is why there is a difference for the Mann prediction (0.87) ---XXX

Typo's etc. Caption Figure 4: Change "divided by the lidar... " into "divided by the variance of the lidar..."

XXX--- Changed to "ratio of the variance of each of the velocity components to that of lidar beam" as suggested---XXX

Page 17, line 11: the reference to Fig 5-left is wrong

XXX--- the "-left" text is now removed ---XXX

Page 17, line 17; the frequencies should be 0.04 and 0.07 Hz.

XXX--- Changed as suggested ---XXX

Page 20, line 15: change into:"...and the sonic-derived u_star to compute L_O and phi_m values"

XXX--- Changed as suggested ---XXX

Page 24, line 31/32 change into: "we make use of"

XXX--- Changed as suggested ---XXX

Page 31, figure 21 right: probably a wrong colour is used for PHI_vw; it should be light blue (instead of black)

XXX--- Thanks for pointing to this very specific issue! We now use the light blue color here ---XXX

There is something peculiar with figure 14: in the left figure 2 data points are visible in the column for sig^2_cup between 6 and 8 m^2/s^2; and 2 data points in the column between 8 and 10. In the right figure it are 4 and 2 data points resp. (In Fig 15 left it are resp 4 and 3 data points and in 15 right 4 and 3

XXX--- Thanks for pointing to this very specific issue as well! In Fig. 14-left, 2 points were just above 10 m2/s2 within the 6—8 m2/s2 bin and 1 within 8—10 m2/s2. The third point within the 8—10 m2/s2 bin in Fig. 14-right is well above 11 m2/s2 together with another point on a lower bin so that it is more complicated to locate (we clarify this in the caption) ---XXX

---

## Author Comment (AC2) · 2 Feb 2017

Dear Referee,

Thank you very much for your comments on the manuscript, which are very helpful and valuable. Herewith our response to your comments in red: XXX--- the response ---XXX
* * *
The characterization of 3-D turbulence from lidar for more accurate quantification of the rotor-disk wind resource and for predicting wind power generation and loads is in critical need, especially as lidar become more prevalent across wind farms. The paper by Pena et al. presents two methodologies for improving our understanding of and measurement techniques for obtaining accurate estimates of atmospheric turbulence across a turbine rotor-disk using forward-looking nacelle-mounted lidar systems. Lidar measurements of the radial-velocity, as well as the velocity spectra, and derived variances, are compared against tower-mounted cup anemometer and sonic anemometer measurements. Results are presented for both pulsed and continuous wave lidar systems and as a function of beam orientation, atmospheric stability, wind speed class, and cone angle. The authors conclude from the results that the use of a central beam and a larger cone angle would improve the accuracy of lidar turbulence measurements.

This topic is of high interest to the wind energy community as research investigating the sizes of turbulence and effects of 3-D turbulence on power generation and fatigue loads is currently being presented by numerous research groups, often offering alternative methodologies for obtaining accurate estimates of turbulence from lidar. I recommend acceptance of the manuscript after revisions, largely to help with the clarity of the results findings.

Major points: The manuscript is currently very long; results are not presented until page 20. I recommend that the authors consider whether the material presented in Section 2: General Background can be shortened. Does this information exist in earlier publications and can be largely cited here instead of explained in detail? The same comment is relevant for Section 3. These sections would be easier for the reader to digest if they were made more concise.

XXX--- We agree that the manuscript is long. The experiment and data analysis were already presented from page 9. We made a reduction of both sections (2 and 3) keeping the background that we believe is basic to understand and reproduce the results of the paper ---XXX

The use of sonic anemometry and cup anemometry is confusing throughout the results section. Please state that the sonic anemometer is 3-D. Only someone familiar with the CSAT3 would be aware of this since it is not mentioned in the text (as far as I can see). I recommend that a discussion is added either to Section 4 or to the Discussion Section which states the measurement differences between all of the instruments. This is briefly mentioned at the end of the manuscript but the point is important. A cup anemometer does not measure the three velocity components; instead mean horizontal velocity and variance is measured. Because of this, the reader is left wondering

why the authors rely so heavily on the cup anemometer measurements for comparison to the lidar estimates of variance. At the very least a discussion needs to be included which outlines the limitation of deriving turbulence measurements from a cup anemometer.

XXX--- We now add 3-D as suggested.  The last paragraph of Section 5.2 contains explanations related to when sonic-derived turbulence measures and when cup-derived turbulence measures are used in the results. We also explained (when presenting the original Fig. 8-right) that adding the v-component to the variance analysis (between cup and sonic) did not change the results (influence of the v-component can be problematic when estimating turbulence with cups). We also showed that there are some issues regarding the sonic spectra. Since the cup-anemometer is closer to the hub height and since the u-statistics of the sonic (means and variances) are systematically lower than those from the cup and the mean wind speeds are also systematically lower than the lidar-derived values (both when reconstructing the beams and when looking at the central beam only), we decided to use the sonic variance as proxy for the u-variance. The first paragraph of the discussion section also provides insights regarding the sonic, cup and lidar measurements. In that section we now include also references to literature giving further insights regarding the three types of instruments ---XXX

The discussion would be strengthened by comparing these results to prior studies that have derived or utilized methodologies for estimating turbulence from lidar. Many of these studies use vertical-profiling or scanning lidar, however they are still relevant. Examples include recent work by J. Newman et al. (2016) whose group examined the accuracy of lidar variance against 3-D sonic anemometry. Lastly, please add additional discussion concerning the motivation for needing better lidar turbulence measurements. This is briefly addressed in the abstract "....useful to predict the loads on a turbine", however the connection between the two is left up to the reader without additional information. Also, please discuss the connection between turbulence and power generation as many recent studies have been published in this area.

XXX--- The last paragraph of the discussion section is now enlarged to take the reviewer's comments into account ---XXX

Minor points: 1. Figure 1. Please connect the ends of the beam lines. It took this reviewer a long time to realize that beam #2 and beam #4 were not in the same plane. Also, why are 13 beams drawn for the CW system? Is this an arbitrary number?

XXX--- The figure is changed as suggested. 13 beams are just arbitrary so we now add "arbitrary" in the text when presenting the CW system ---XXX

2. P 5 Line 20. Please be more specific. Which frequencies does the lidar average out? This includes "most eddies" of what size?

XXX--- These sentences are now moved to the first paragraph of Section 3.2 and reference to Figs. 2 and 3 is made, where it is clear why we write "most eddies" ---XXX

3. P 6 Line 3. Elaborate on why the sonic u-spectrum is considered "ideal".

XXX--- We now add text that explains our wording, as suggested ---XXX

4. P 6 Line 6-7. "...and upward pointing beam spectra is smaller than the differences between ordinary velocity-component spectra..." What are ordinary velocity-component spectra?

XXX--- We now change "ordinary" to "u, v and w" ---XXX

5. Page 6 and throughout, Please comment on what Zr/L represents. If I understand correctly Zr is constant for a particular lidar, so this ratio is a function of atmospheric stability?

XXX--- $Z\_r$ increases with focused distance for the CW lidar. As suggested, we now add a sentence stating what this ratio indicates ---XXX

6. Page 6 L 31. Aren't the two stress fluxes small because of the applied coordinate rotation?

XXX--- Yes. This is because we make sure the u-fluctuations are in the mean wind direction. We now add this to the sentence ---XXX

7. Page 12 L 12-13. The last part of this sentence "...., five days were used to measure with the ZephIR..." is confusing and does not warrant inclusion. Better to leave this info out.

XXX--- Removed as suggested ---XXX

8. Page 13 L 30. The 5-s and 18-s time period were chosen to mimic the lidar sampling frequencies, right?

XXX--- Yes. This info is now added ---XXX

9 Figure 7, top right. What are the dots with a value of 1 along the y-axis? Are they blade interference? Better to remove these points from the figure and use the same y-axis for the left and right panels.

XXX--- Yes, they are blade interference. We thought that it was nice that the reader could see that we are able to filter these values but those are also shown in the bottom right plot so your suggestion is now taken ---XXX

10. Page 15 Line 5. Please comment on the appropriateness of the logarithmic profile for the site here.

%%%--- The log profile is here used simply to get an indication of the increase of wind sped due to the height difference between the cup and the sonic. Based on the comment we now add that the estimation assuming the log profile is made between the instrument's heights. For the interest of the reviewer the profile that best fits the measurements using the cups at 78, 57 and 33 m is slightly stable: the bias estimation by assuming such a profile is 0.8% ---XXX

11. Figure 8. Left panel. Is this not showing evidence of overspeeding by the cup anemometer?

XXX--- As stated in the response to the second comment, when only looking at the mean, the sonic shows biases with both the cup and lidars (and to the central beam of the pulsed lidar). Overspeeding, in the most extreme cases, can perhaps cause a 10% bias (see Bush and Kristensen, 1976) but the conditions at the site are far from extreme ---XXX

12. Page 15 L5-11. This section needs to be discussed in terms of instrument measurement technique differences and common errors associated with each measurement device. Is it not surprising that the cup anemometer is measuring higher variance than the sonic since it may be contaminated by the w component. It needs to be clear that horizontal variance from a sonic and cup anemometer are not expected to be equal, there will be a bias. Is this total horizontal or u?

XXX--- This is also related to the previous and second comment of the reviewer. The first sentence in Section 5.2 states that this is horizontal wind speed comparison. The biggest contribution to cup and sonic variances are due to v-contamination and as stated in those lines we performed the comparison with and without v (for the variance) and found the same results ---XXX

13. Page 16 L 5. The reader may not know which scales are included in the inertial subrange. Please state.

XXX--- Added as suggested ---XXX

14. Page 15 L 13. Doesn't local isotropy assume neutral conditions? If the conditions are stable, wouldn't this explain why the w spectrum is also lower than the u spectrum?

XXX--- Local isotropy does not assume neutral conditions. Even in very stable conditions the w-spectrum should be above the u-spectrum within the inertial subrange ---XXX

15. Figure 10 and text below. The sonic and cup anemometers appear to not suffer from noise at the high frequency end of the spectra. So why was the same noise filter applied to these data, especially since it "distorted the shape" of the sonic u-spectrum?

XXX--- As stated in the text around Fig. 10 the noise filter is applied to the Avent spectra and tested for the 18-s sonic u-spectrum to find out if could be used for the 18-s Zephir spectra. We

now add text along these lines that for the results thereafter presented the the noise filter is only applied to the Avent spectra ---XXX

16. Figure 12. Please list the date periods that these lines correspond to. Also the stability conditions for each.

XXX--- This figure is only shown for illustration of the types of Doppler spectra we were measuring ---XXX

17. Page 20 L 25-30. Instead of using (1/L)-1, why not discuss z/L here since these values are in the table. Plus, isn't (1/L)-1 just L?

XXX--- We now discuss the values in terms of $z/L_O$ as suggested ---XXX

18. Figure 13 Left panel. Why is it assumed that sigma_u/U is constant as a function of wind speed?

XXX--- We do not make this assumption for plotting the data. In the figure we included the prediction (the line) using neutral surface layer theory (which shows that turbulence intensity is constant with wind speed) for illustration only (as described in the text) ---XXX

19 Table 1. I don't think mean values of z/L are meaningful, especially since the mean values you list are well above all of your stability classes. Try using median values instead.

XXX--- This is true for the speed classes since $L_O$ values can be highly fluctuating. We use the median specifically for these classes as suggested ---XXX

20. Page 28 Line 22. Please talk about the fundamental differences in way the instruments measure velocity and turbulence. It is more than "due to the way they probe the atmosphere".

XXX--- See our response to the second major point and related minor comments ---XXX

21. Page 30 Line 10. How do you get uw and w variance from a cup anemometer?

XXX--- We now include ", when compared to those from the sonic anemometer," to clarify the sentence ---XXX

22. Page 32 Line 14. The authors conclude that a larger cone angle would improve estimates of turbulence, but doesn't this make the assumptions about flow homogeneity across the scanning cone less valid

XXX--- This is right. We now add text to the sentence regarding this issue ---XXX

References:

Bush N and Kristensen (1976) Cup anemometer overspeeding. J. Appl. Meteorol., 15, 1328—1332